# Eye-brain connections revealed by multimodal retinal and brain imaging genetics

Bingxin Zhao [1,2,3,4,5,6,7] ✉, Yujue Li[2], Zirui Fan[1], Zhenyi Wu[2], Juan Shu[2], Xiaochen Yang[2], Yilin Yang[1], Xifeng Wang[8], Bingxuan Li [9], Xiyao Wang[9], Carlos Copana [2], Yue Yang [8], Jinjie Lin[10], Yun Li [8,11], Jason L. Stein [11,12], Joan M. O'Brien[13,14], Tengfei Li [15,16] & Hongtu Zhu [8,11,17,18] ✉

The retina, an anatomical extension of the brain, forms physiological connections with the visual cortex of the brain. Although retinal structures offer a unique opportunity to assess brain disorders, their relationship to brain structure and function is not well understood. In this study, we conducted a systematic cross-organ genetic architecture analysis of eye-brain connections using retinal and brain imaging endophenotypes. We identified novel phenotypic and genetic links between retinal imaging biomarkers and brain structure and function measures from multimodal magnetic resonance imaging (MRI), with many associations involving the primary visual cortex and visual pathways. Retinal imaging biomarkers shared genetic influences with brain diseases and complex traits in 65 genomic regions, with 18 showing genetic overlap with brain MRI traits. Mendelian randomization suggests bidirectional genetic causal links between retinal structures and neurological and neuropsychiatric disorders, such as Alzheimer's disease. Overall, our findings reveal the genetic basis for eye-brain connections, suggesting that retinal images can help uncover genetic risk factors for brain disorders and disease-related changes in intracranial structure and function.

The retina, an important component of the central nervous system that can be non-invasively visualized through retinal imaging, plays a critical role in the visual pathway. It connects synaptically to the visual cortex through the optic nerve, thalamus, and optic radiations. The retina and brain share anatomical, physiological, and embryological similarities in cell types, vasculature, and immune responses[1]. During the third week of gestation, the eye develops from the forebrain[2], with the retina originating from the diencephalon, which later becomes the thalamus. This developmental connection links the retina to specific brain regions. As a result, the retina serves as a unique window into brain structure/function[1,3] and various disorders[4], such as Alzheimer's disease[5–8], Parkinson's disease[9], stroke[10,11], cerebral small vessel disease[12], schizophrenia[13], cognitive decline[11,14,15], and many others. Retinal neurodegeneration, for example, is strongly associated with amyloid β (Aβ) burdens in Alzheimer's disease and has been widely studied as an easily accessible biomarker for identifying individuals at high risk of developing Alzheimer's disease or those with preclinical Alzheimer's disease[6,8,16–19]. Additionally, retinal abnormalities have been frequently reported in Parkinson's disease, with animal models showing similar molecular mechanisms underlying Parkinson's disease pathology and neurodegeneration in Parkinsonian eyes[9]. Despite these connections, limited knowledge exists on shared genetic effects underlying eye-brain relationships and parallel pathological changes between the two organs, except for a few disease pairs like primary open-angle glaucoma and Alzheimer's disease[20].

The retina and brain images offer well-defined clinical endophenotypes for eye and brain disorders. Widely used retinal imaging modalities include color fundus photography and optical coherence tomography (OCT). Retinal images serve as the gold standard for screening age-related macular degeneration[21], diabetic retinopathy[22], and other retinal pathologies. These images provide a colorful view of the eye's posterior, encompassing the retina, optic nerve head, and retinal vasculature. Retinal OCT imaging presents a high-resolution cross-sectional view of the retina[23]. In neurological conditions, OCT imaging facilitates the evaluation of retinal layer thickness and structural alterations due to neuronal and retinal glial cell modifications[24]. In addition, magnetic resonance imaging (MRI) captures brain structure and function, resulting in various clinical applications for neurological and neuropsychiatric disorders[25]. Recent large-scale genome-wide association studies (GWAS) have demonstrated the heritability of both retinal imaging biomarkers[26–34] and brain MRI traits[35–42], with genetic influences from common genetic variants identified in numerous genomic regions. As anticipated, genetic overlaps were discovered between retinal imaging traits and eye disorders, such as optic nerve head cupping and glaucoma[31], as well as between brain MRI traits and brain disorders, like functional connectivity of the visual network and Alzheimer's disease[40]. However, few studies have used imaging genetics to examine brain health from a retinal perspective. A comprehensive cross-organ analysis of retinal and brain imaging traits could potentially offer an opportunity to identify retinal imaging biomarkers for brain disorders and to uncover the genetic basis for eye-brain connections.

We explored the genetic relationship between the eye and brain by analyzing multimodal retinal and brain imaging traits from the UK Biobank (UKB) study[43]. The majority of our study's cohort consists of healthy subjects, with an overall low prevalence of ocular or brain disorders (Supplementary Note). We examined a total of 156 retinal imaging traits, including 46 derived from OCT images and 110 from fundus photographs. The OCT-derived measures, including retinal thickness across layers[44,45] (such as the retinal nerve fiber layer [RNFL], inner nuclear layer [INL], and the ganglion cell and inner plexiform layer [GCIPL]) and vertical cup-to-disc ratio[30], were already available in the UKB database. For fundus images, we used 11 different pre-trained transfer learning[32] models built from ImageNet[46]. We considered the top 10 principal components (PCs) in each deep transfer learning model, accounting for an average of 70.71% variance (range = [50.58%, 95.84%]) in the final layer, resulting in 110 fundus image features (11 × 10). These deep learning-based image embeddings and low-dimensional representations contain eye-specific biological information not found in standard eye measurements[32]. We conducted GWAS for these 156 retinal imaging traits and assessed their genetic correlations with 458 imaging traits from three primary brain MRI modalities: (1) 101 regional brain volumes[36] and 63 cortical thickness traits[47] from structural MRI; (2) 110 diffusion tensor imaging (DTI) parameters from diffusion MRI[38]; and (3) 92 functional connectivity and activity (or amplitude) traits from resting-state and task-based functional MRI (fMRI)[40]. Further details on these retinal and brain imaging data, such as the specific retinal layers and brain regions, are provided in the "Methods" section and Supplementary Data 1. Figure 1 provides an overview of the study design and data analysis. GWAS summary statistics for retinal imaging traits and our data analysis results will be accessible through the eye imaging genetics knowledge portal (Eye-KP) at https://www.eyekp.org/.

## Results

### Phenotypic multimodal eye-brain connections

We examined phenotypic associations between 156 retinal imaging traits and 458 brain MRI traits after adjusting for a wide variety of vascular risk factors[3] and imaging confounders[35], as well as body size, age, and sex effects (see the "Methods" section for the complete list of adjusted covariates). For discovery, we analyzed data of UKB white British individuals (average $n = 6454$ across different modalities). At the false discovery rate (FDR) level of 5% (by the Benjamini–Hochberg procedure, $P < 4.37 \times 10^{-4}$, 156 × 458 tests), we identified 625 associations (Figs. 2A and S1), 135 of which were replicated in a hold-out independent validation dataset (average $n = 959$) with concordant association signs (Fig. S2). Among the 625 associations, 121 further survived the conservative Bonferroni significance level ($P < 6.99 \times 10^{-7}$, 156 × 458 tests), and 66 can be replicated in the same hold-out independent dataset. These significant results were mainly related to multiple brain structural modalities, including regional brain volumes, cortical thickness, and DTI parameters. They were broadly related to both OCT measures and fundus image features (Supplementary Data 2). Below we summarized the patterns of associations that have been replicated.

Thickness measures of the macular[45], RNFL[44], GCIPL[44], and INL[44] were positively associated with volumes of multiple brain cortical and subcortical structures, including the pericalcarine, thalamus, pallidum, and putamen (Figs. 2B, C and S3). The pericalcarine is the location where the primary visual cortex (V1) concentrates, and we found consistent positive associations between regional brain volumes of the pericalcarine and the RNFL, GCIPL, and macular thickness ($\beta > 0.052$, $P < 5.90 \times 10^{-5}$). We also observed positive associations with brain structures in the dorsal and ventral visual pathways extending from the primary visual cortex, such as the cuneus ($\beta > 0.057$, $P < 1.08 \times 10^{-4}$). The thalamus and macular both originate from the diencephalon, and positive associations between regional brain volumes of the thalamus and macular thickness were found ($\beta > 0.120$, $P < 1.24 \times 10^{-8}$), emphasizing their developmental origins. Negative associations between retinal layer thickness and enlargement of the lateral ventricles were also detected. The left and right hemispheres of the brain demonstrated consistent associations with retinal imaging traits. For example, the left and right brain thalamus volumes were significantly correlated with the thicknesses of the macular and GCIPL in both eyes ($\beta > 0.120$, $P < 1.24 \times 10^{-8}$). These regional brain volume traits were extracted using advanced normalization tools (ANTs)[48], we additionally conducted further analyses with traits of visual cortical regions generated from Freesurfer[49], which yielded consistent findings (Supplementary Note and Fig. S4). The GCIPL thickness was also positively associated with global and regional brain cortical thickness measures, including the primary visual cortex (the pericalcarine, $\beta = 0.048$, $P = 7.74 \times 10^{-5}$). The top two regions with the strongest links were the precuneus, which is in the dorsal visual pathway ($\beta = 0.073$, $P = 2.33 \times 10^{-8}$), and the fusiform, which is in the ventral visual pathway ($\beta = 0.064$, $P = 4.22 \times 10^{-7}$, Figs. 2D and S3).

Moreover, thicknesses of the macular, RNFL, and GCIPL consistently exhibited positive associations with the fractional anisotropy (FA) of multiple white matter tracts, including those related to the visual pathway (Figs. 2E, F and S5). Retinal thinning is linked to various eye diseases[50], and previous studies have shown that the thinning of the RNFL and GCIPL is associated with cerebrovascular diseases[51] and early-stage Alzheimer's disease[52]. These results suggest a parallel relationship between retinal and brain health, along with changes in brain white matter that may be connected to both. The strongest associations were observed between the left eye's GCIPL thickness and the mean FA of brain posterior thalamic radiation, sagittal stratum, and fornix-stria terminalis tracts ($\beta > 0.142$, $P < 1.91 \times 10^{-23}$). The posterior thalamic radiation overlaps with the optic radiation in the visual pathway, which connects the lateral geniculate nucleus to the primary visual cortex and transmits visual input from the eye. Similar associations between fundus image features and DTI parameters were found, although they were weaker than those for retinal thickness traits (Fig. S5).

We conducted separate analyses for females and males to examine sex-specific patterns (average $n = 3338$ and 3150,

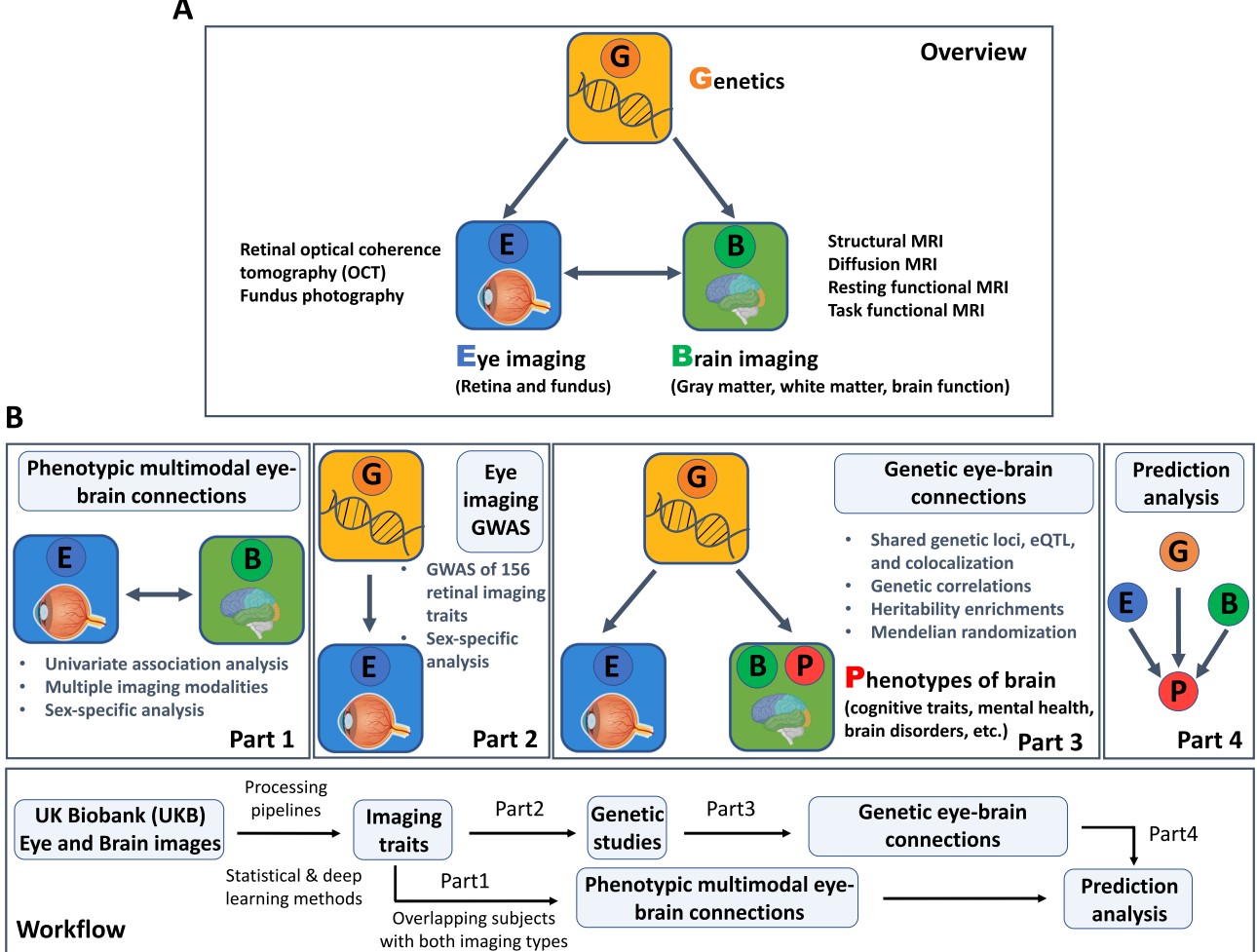

**Fig. 1 | Study overview and workflow. A** An overview of the study design. We used multimodal retinal and brain imaging data to understand the phenotypic and genetic connections between the brain and the eye. We considered multiple brain magnetic resonance imaging (MRI) modalities, including structural MRI, diffusion MRI, resting-state functional MRI (fMRI), and task-based fMRI. For the eye, we used traits derived from retinal optical coherence tomography (OCT) and extracted from fundus retinal images using pre-trained transfer learning models. **B** A brief description of the overall workflow and major analyses in each part. This figure is created with BioRender.com released under a Creative Commons Attribution-NonCommercial-NoDerivs 4.0 International license.

respectively). At the FDR 5% level ($P < 4.37 \times 10^{-4}$), 53 associations were identified in both females and males, with an additional 191 additional associations found in females, and 62 more found only in males. The extra associations found in analyses including only females or males were primarily related to fundus image traits. Specifically, the female analysis showed more significant associations with DTI parameters, while the male analysis revealed more significant associations with cortical thickness measures (Figs. S6–S8). For OCT measures, males and females exhibited similar eye-brain association patterns, although the number of significant pairs that survived multiple testing adjustments varied between the two samples. For example, the mean FA of the fornix-stria terminalis, posterior thalamic radiation, and sagittal stratum tracts was associated with the thickness of RNFL, GCIPL, and macula in both males and females, with more significant pairs being identified in females (Fig. S9). These retinal thickness traits were also consistently associated with volumes of the pericalcarine, thalamus, and accumbens regions in sex-specific analyses (Fig. S10). We performed additional sensitivity analysis to account for the ocular and brain disorders as covariates. We found consistent patterns with those observed in our main analysis, more details can be found in the Supplementary Note (Figs. S11–12). In summary, although only a relatively small percentage of subjects in the UKB study had both brain and retinal imaging

data, we discovered that retinal imaging biomarkers, such as the thickness of different retinal layers, were associated with smaller brain volumes, reduced cortical thickness, and weaker white matter structural connections in the brain. Many retina-related brain structural variations were observed in the primary visual cortex and structures in the visual pathways.

## GWAS for 156 retinal imaging traits

We analyzed data from UKB individuals of white British ancestry[53], estimating phenotypic variance explained by single nucleotide polymorphisms (SNPs) for the 156 retinal imaging traits (average $n = 60,748$). The average SNP-based heritability ($h^2$) was 42.21% for the 46 OCT measures ($h^2$ range = (19.28%, 68.28%)), all of which were significant at a 5% FDR level (Fig. 3, Fig. S13, and Supplementary Data 3). Out of the 110 fundus image traits, 90 were significant at the 5% FDR level, and the mean $h^2$ was 19.27%, ranging from 4.06% to 42.75%. In each of the 11 transfer learning models, at least seven out of the 10 PCs showed significant $h^2$. Additionally, we estimated $h^2$ separately for females and males, and the results were highly consistent between the two sexes, with a mean $h^2$ of 24.83% in females and 23.33% in males (correlation = 0.972, $P = 0.457$, Fig. S14).

We conducted a GWAS using the same white British cohort to investigate the genetic architecture of the 156 retinal imaging traits

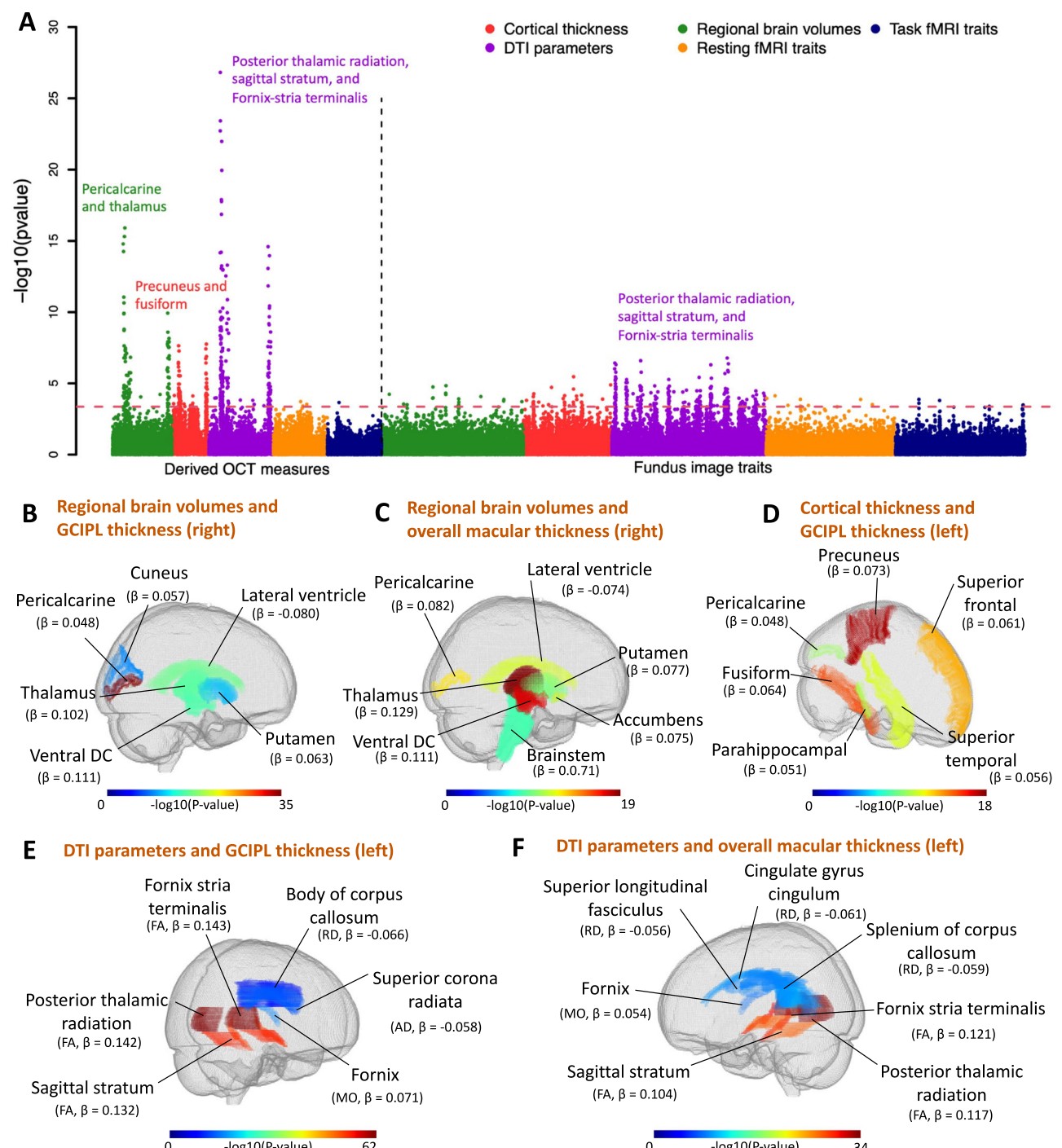

**Fig. 2 | Phenotypic eye-brain imaging associations. A** This figure shows the −log10(*p*-value) for testing associations between 156 retinal imaging traits (46 derived OCT measures and 110 fundus image traits) and 458 brain MRI traits, comprising 101 regional brain volumes, 63 cortical thickness traits, 110 diffusion tensor imaging (DTI) parameters, 92 resting fMRI traits, and 92 task fMRI traits. Supplementary Data 1 provides further information on these imaging traits. The red dashed horizontal line signifies the Benjamini–Hochberg FDR 5% significance level (raw *P* < 4.37 × 10⁻⁴). Each brain imaging modality is marked with a different color, and the brain structures with the strongest associations in each modality are labeled. **B**, **C** The location of brain regions with volumes significantly associated with **B** the thickness of the ganglion cell and inner plexiform layer (GCIPL, right eye) and **C** the overall thickness of the macula (right eye). **D** The location of cortical brain regions exhibiting a significant association between their thickness and the thickness of GCIPL (left eye). **E**, **F** The location of white matter tracts with DTI parameters associated with **E** the thickness of GCIPL (left eye) and **F** the overall thickness of the macula (left eye). AD axial diffusivity, RD radial diffusivity, MO mode of anisotropy, FA fractional anisotropy. For all associations, we used two-sided *t*-tests and adjusted for multiple comparisons. Matplotlib[157] contributed to the creation of this figure.

(average *n* = 60,748). QQ and Manhattan plots can be viewed on our server (http://165.227.78.169:443/) created using PheWeb[54]. Linkage disequilibrium score regression (LDSC) intercepts[55] were all close to one, suggesting no confounding factors resulted in genomic inflation of test statistics (average = 1.004, range = (0.974, 1.031)). Sensitivity

analysis, which additionally included ocular and brain disorders as covariates, yielded highly consistent GWAS results (Supplementary Note). Applying a stringent GWAS significance level of 3.20 × 10⁻¹⁰ (which is 5 × 10⁻⁸/156, accounting for Bonferroni-type adjustment for 156 retinal imaging traits), we identified independent

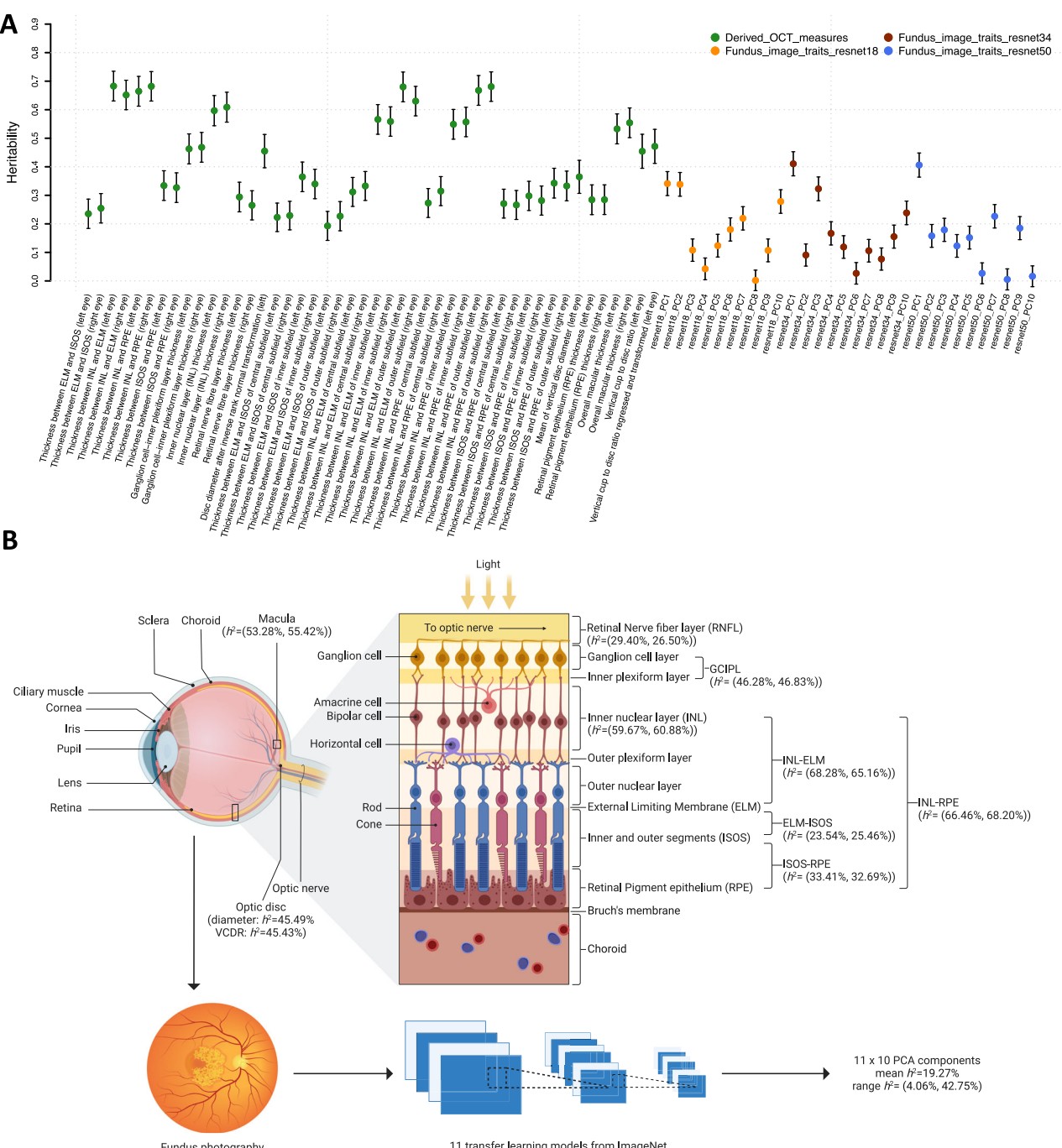

**Fig. 3 | Heritability of selected retinal imaging traits. A** SNP heritability of the 46 derived OCT measures and 30 fundus image traits from three selected transfer learning models. The *x*-axis displays the retinal imaging traits, see Supplementary Data 1 for more information on these traits and Fig. S13 for the heritability of more fundus image traits. The error bars represent 95% confidence intervals based on the standard errors of the point estimates and the assumption of normality. **B** We show the relative locations of 18 selected OCT measures and their SNP heritability ($h^2$), including the thickness of macula, RNFL, GCIPL, INL, INL-ELM, ELM-ISOS, ISOS-RPE, and INL-RPE, optic disc diameter, and VCDR (vertical cup-to-disk-ratio). **B** Created with BioRender.com released under a Creative Commons Attribution-NonCommercial-NoDerivs 4.0 International license. All values are represented as $h^2 =$ (left eye, right eye) except for the optic disc, where only the left eye data is available. In the lower panel, we also illustrate the process of acquiring fundus imaging traits from fundus photography and the heritability statistics of the 110 resulting traits. ELM external limiting membrane, ISOS inner and outer photoreceptor segments, INL inner nuclear layer, RPE retinal pigment epithelium, RNFL retinal nerve fiber layer, GCIPL ganglion cell-inner plexiform layer.

(linkage disequilibrium [LD] $r^2 < 0.1$) significant genetic associations in 258 genomic regions (cytogenetic bands). We found significant associations for all 46 OCT measures and 91 of the 110 fundus image traits (Fig. S15 and Supplementary Data 4). Furthermore, we estimated these significant genetic effects separately for males and females in a sex-specific analysis. The genetic effects showed a high level of consistency in both sexes (correlation = 0.975, $P = 0.76$, Fig. S16).

We validated our GWAS findings using independent European and non-European datasets ("Methods" section). First, we conducted a GWAS of the 156 retinal imaging traits using the UKB participants of European ancestry but not of British origin (average $n = 5320$). Among

the 4329 identified independent (LD $r^2 < 0.1$) image-variant associations in 258 genomic regions, 1630 (37.65%, in 162 genomic regions) passed the 5% FDR significance level in this European validation GWAS, and 2210 (51.05%, in 189 regions) were significant at the nominal significance level (0.05) (Fig. S17 and Supplementary Data 5). Most of the significant genetic effects (2207/2210, in 188 regions) had concordant directions in the two independent GWAS, with a correlation of 0.958 for their genetic effects (Fig. S18). Among the 188 replicated genomic regions, 146 were associated with OCT measures, and 103 were associated with fundus image traits, suggesting the high generalizability of our GWAS findings in European samples. Next, we performed a validation GWAS on non-European UKB subjects (average $n = 6490$). We found that 25.18% (1090/4329, in 142 regions) associations were significant at the nominal significance level, with most of them (1068/1090, in 140 regions) having the same genetic effect directions as the discovery GWAS (Fig. S19). In non-European validation analysis, 107 replicated regions were observed for OCT measures and 51 for fundus image traits. We also performed the replications analysis separately for UKB subjects of Asian and Black ancestries (Supplementary Note).

We developed polygenic risk scores (PRS) via PRS-CS[56] to assess the out-of-sample prediction performance of our discovery GWAS results ("Methods" section). The PRS for 133 of the 156 retinal imaging traits were significant at a 5% FDR level ($P$ range = $(6.90 \times 10^{-98}, 3.82 \times 10^{-2})$, Supplementary Data 6 and Fig. S20), with the mean incremental R-squared being 2.51% (S.E. = 2.29%). A total of 22 traits had R-squared greater than 5%. The highest prediction accuracy was observed for traits related to the INL[44], such as the thickness from the INL to the retinal pigment epithelium (RPE) (R-squared = 9.46% and 8.43% for right and left eyes, respectively) and the thickness between the INL to the external limiting membrane (ELM) (R-squared = 7.68% and 7.20% for right and left eyes, respectively). To evaluate the transferability of our GWAS findings, we examined the PRS performance on non-European UKB subjects as well. We discovered that 101 retinal imaging PRS had significant prediction performance in the non-European UKB dataset at a 5% FDR level ($P$ range = $(2.77 \times 10^{-65}, 2.88 \times 10^{-2})$). The average incremental R-squared of these significant PRS was 1.39% (S.E. = 1.45%), which was significantly lower than their corresponding performance in the European dataset ($P = 7.89 \times 10^{-9}$). These results demonstrate the capability of our GWAS summary statistics in out-of-sample analyses and illustrate the challenge of cross-population genetic prediction.

## Genetic underpinnings of eye-brain connections in 65 genomic loci

We investigated eye-brain genetic pleiotropy in 188 replicated genomic regions of retinal imaging traits that exhibited concordant genetic effect directions in the discovery and validation GWAS. First, for the retinal imaging-significant genetic variants and those in LD with them ($r^2 \geq 0.6$), we symmetrically searched for GWAS signals associated with brain MRI traits[36,38,40,47]. Second, we performed association lookups in the NHGRI-EBI GWAS catalog[57] to identify shared genetic influences between retinal imaging traits and brain-related complex traits and diseases ("Methods" section). In 65 of these 188 genomic regions, we discovered genetic overlaps between retinal imaging traits and brain phenotypes, with 47 of them also being linked to various eye traits and conditions, such as glaucoma[58], refractive error[59], advanced age-related macular degeneration[60], and cataracts[61]. Specifically, we identified genetic pleiotropy for a wide range of brain traits and disorders, including stroke, Parkinson's disease, Alzheimer's Disease, glioma/glioblastoma, neuropsychiatric disorders, migraine, mental health, and cognitive traits (Fig. 4 and Supplementary Data 7). Shared genetic influences were also detected in 18 regions with different brain MRI modalities, including 10 regions with regional brain volumes[36], 9 regions with DTI parameters[38], 3 regions with cortical thickness traits[47], and 2 regions with resting fMRI traits[40] (Fig. S21). Using Bayesian

colocalization analysis[62], we examined whether common causal genetic variants underlie the overlapping genetic signals between retinal structures and brain phenotypes (posterior probability of the shared causal variant hypothesis [PPH4] >0.8[62,63]). In addition, we found that many retinal imaging-significant genetic variants were expression quantitative trait loci (eQTLs) reported in large-scale eQTL studies of brain tissues[64]. Our results are summarized in Supplementary Data 8, with selected eye-brain trait pairs displayed in Figs. 5, 6 and S22–S82. We provide more details below for each brain MRI modality and major brain phenotype category.

We found genetic overlaps between brain volumetric measures and retinal structures in 10 genomic loci (LD $r^2 \geq 0.6$, Figs. 5A and S22–30). For example, shared genetic components were discovered between cerebrospinal fluid (CSF) volume and vertical cup-to-disc ratio[65] in 11q24.3 (Fig. 5A). The retinal index variant rs4937515, which is an eQTL of *ADAMTS8* in brain tissues[64], showed strong evidence of shared causal genetic variants between the two traits (PPH4 = 0.997). This variant was also in LD ($r^2 \geq 0.6$)[66] with known glaucoma risk variants (index variant rs2875238)[58]. CSF is the primary fluid within the central nervous system, and its pressure plays a well-established role in glaucoma and other ophthalmic diseases[67,68]. Our findings reinforce the genetic connections between CSF and eye disorders. Colocalizations between retinal imaging traits and brain volumes were also observed in several other regions, such as 8q23.1 (e.g., right thalamus), 22q13.1 (e.g., left lateral ventricle), 17q24.2 (e.g., left caudal anterior cingulate), 6q25.1 (e.g., right hippocampus), and 7q22.1 (e.g., right accumbens area). As the first stop of sensory processing in the visual system, the thalamus plays a fundamental role in the visual pathway[69]. We found that multiple retinal imaging traits, including the thickness of INL-RPE, INL-ELM, and GCIPL, had genetic overlaps with both left and right thalamus volumes in 8q23.1 and 2q24.3. In these brain volume-associated regions, retinal imaging traits were also linked to (LD $r^2 \geq 0.6$) schizophrenia, major depressive disorder, neuroticism, and cognitive traits. In addition, we found genetic overlaps between retinal structures and cortical thickness traits in 3 loci (17q21.31, 8p23.1, and 1q21.3). Several brain structures in the visual pathways were involved in these overlaps, including the precentral, supramarginal, fusiform, and precuneus regions (Fig. 5B). In summary, we identify locus-specific genetic overlaps between the thickness of different retinal layers and the morphometry of multiple brain regions. These brain regions play crucial roles in cognitive functions and are affected by various brain disorders.

Retinal structures demonstrated genetic pleiotropy with brain structural and functional connectivity. Retinal imaging traits and DTI parameters shared genetic influences in 9 genomic regions, with strong evidence of colocalization in 7 of these regions (Figs. 5C and S31–38). For example, overall macular thickness[45] and the mode of anisotropy (MO)[70] of the inferior fronto-occipital fasciculus tract exhibited common causal genetic variants in 17q24.2 (PPH4 = 0.816, Fig. 5C). The inferior fronto-occipital fasciculus is a long associative white matter tract connecting various brain areas and involved in multiple functions. Thinner retinal layers were closely associated with reduced volume and poorer microstructural integrity of the brain's white matter[71]. These findings offer genetic insights into the connections between the retina and white matter. Additionally, both retinal imaging traits and DTI parameters had genetic overlaps with cognitive traits (e.g., intelligence in 22q13.1), psychiatric disorders (e.g., in 14q24.3), and eye disorders (e.g., advanced age-related macular degeneration in 17q25.3). We also found shared genetic influences between functional connectivity of resting fMRI and retinal imaging traits in 11q13.3 and 17q21.31 (Figs. 5D and S39).

Many genomic regions associated with retinal imaging traits have been linked to brain-related complex traits and diseases in previous GWAS. In the 6q14.2, 6q21, 13q14.2, 15q26.1, and 16q22.1 regions, the thickness of different retinal layers was in LD ($r^2 \geq 0.6$) with

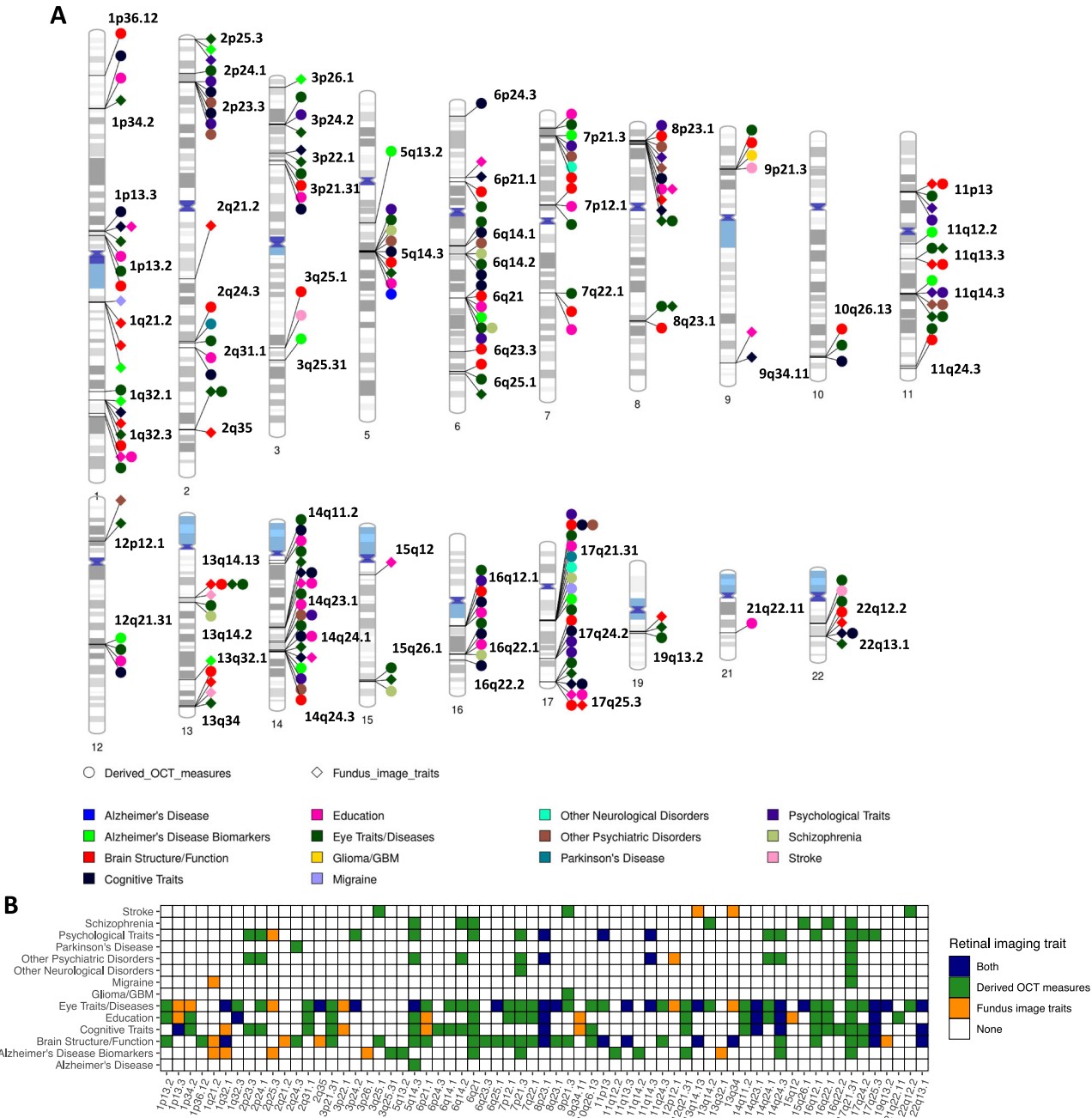

**Fig. 4 | Genomic loci associated with both eye imaging traits and brain-related complex traits and diseases. A** Ideogram displaying genomic regions (names are in black) influencing both retinal imaging traits and brain-related complex traits and diseases, including phenotypes reported on the NHGRI-EBI GWAS catalog (https://www.ebi.ac.uk/gwas/) and brain MRI traits available on BIG-KP (https://bigkp.org/). Each category of brain phenotypes is marked with a different color, and distinct shapes are used for OCT measures and fundus image traits. **B** Table summary, where the *x*-axis represents the genomic regions and the *y*-axis shows the category of brain phenotypes. Derived OCT measures and fundus image traits are indicated with different colors, while a third color is used when both are observed in the same locus.

schizophrenia[72–74] (Figs. 6A and S40–S43). For example, the INL thickness had shared causal genetic variants with schizophrenia (PPH4 = 0.952). The retinal index variant (rs7752421) was an eQTL of *SNAP91* in human brain tissues[64], affecting gene expression levels in the brain. In excitatory neurons, synaptic defects are increasingly associated with schizophrenia, and altered expression of *SNAP91* has been observed to impact synaptic development[75]. Since schizophrenia patients often report changes in visual perception, OCT measures of the retinal structure have gained more attention in schizophrenia research[76]. The genetic links identified in our analysis support the use of retinal layer assessments as potential biomarkers for schizophrenia.

Retinal structures were also in LD ($r^2 \geq 0.6$) with other neuropsychiatric disorders and mental health traits, such as bipolar disorder[77], anxiety[78], depressive symptoms[79], neuroticism[80], subjective well-being[81], and risk-taking tendency[82] (Figs. S44–S52). For example, the thickness of various retinal layers shared genetic influences with neuroticism in multiple regions (Figs. S44–S50).

In addition, 23 genomic regions were associated with cognitive traits (such as intelligence[83], cognitive performance[84], general cognitive ability[85], and reaction time[86], Figs. S53–S60) and/or educational attainment[87] (Figs. 6B and S61–S75). Previous research has reported that retinal layer thickness could serve as a prognostic biomarker of

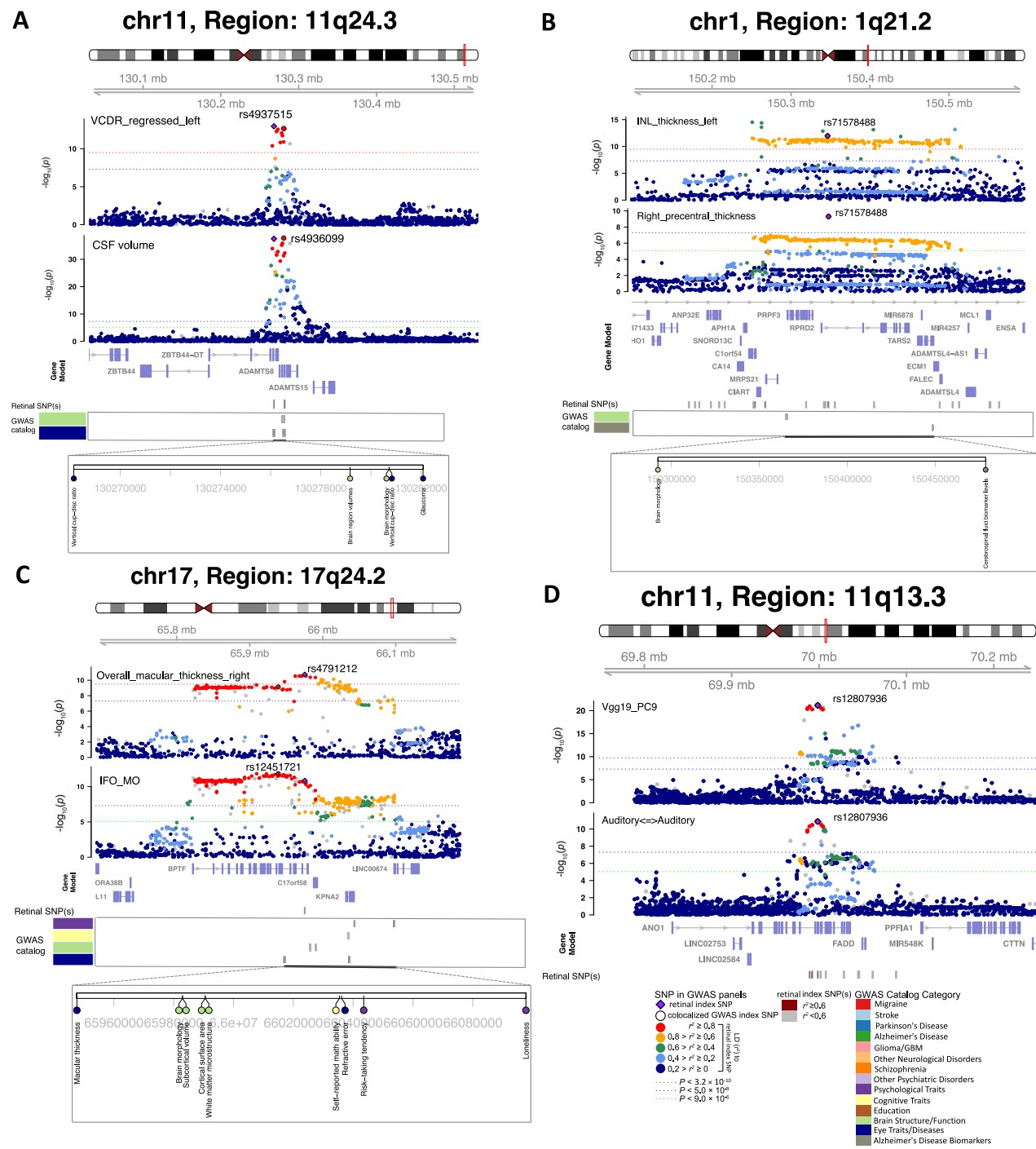

**Fig. 5 | Selected genetic loci that were associated with both eye and brain imaging traits. A** In 11q24.3, shared genetic influences were observed between the vertical cup-to-disc ratio (regressed on disc diameter, left eye, VCDR_regressed_left, index variant rs4937515) and the cerebrospinal fluid volume (CSF volume, index variant rs4936099). Bayesian colocalization analysis suggested the shared causal variant between the two traits (posterior probability PPH4 = 0.997). **B** In 1q21.2, shared genetic influences were observed between the inner nuclear layer (INL) thickness (left eye, INL_thickness_left) and the cortical thickness of the right precentral brain region (Right_precentral_thickness, shared index variant rs71578488, PPH4 = 0.562). In this region, the INL_thickness_left was also in LD ($r^2 \geq 0.6$) with cerebrospinal fluid biomarker levels. **C** In 17q24.2, shared genetic influences were observed between the overall macular thickness (right eye, overall_macular_thickness_right, index variant rs4791212) and the mean MO of the inferior fronto-occipital fasciculus (IFO_MO, index variant rs12451721, PPH4 = 0.963). We also observed genetic overlaps (LD $r^2 \geq 0.6$) with self-reported math ability, risk-taking tendency, and loneliness. **D** In 11q13.3, shared genetic influences were observed between the ninth PC of the Vgg19 model on fundus image (Vgg19_PC9) and the functional connectivity within the auditory network (Auditory < ≥Auditory, shared index variant rs12807936, PPH4 = 0.994).

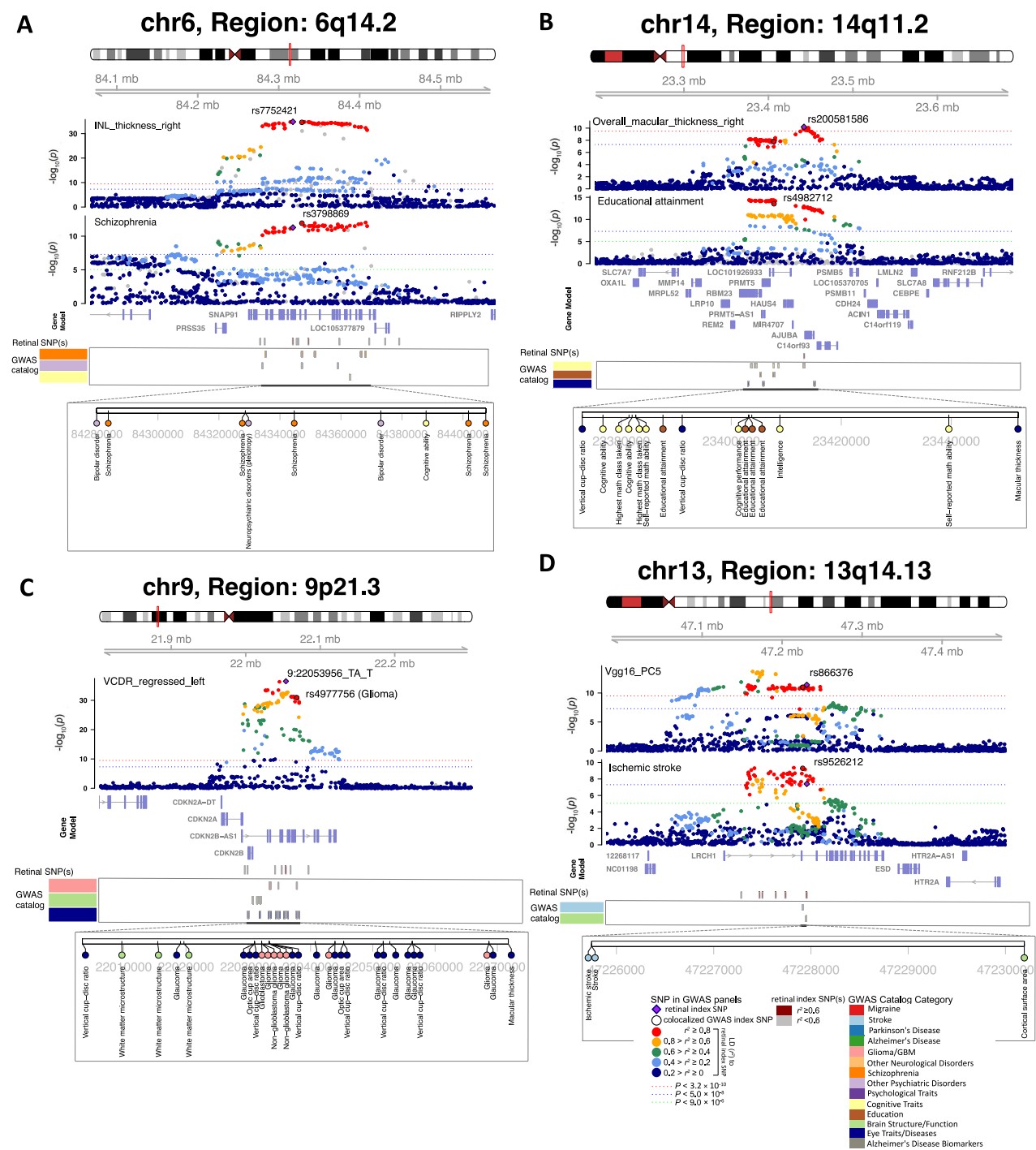

**Fig. 6 | Selected genetic loci that were associated with both eye and brain-related complex traits and disorders.** A In 6q14.2, shared genetic influences were observed between the inner nuclear layer (INL) thickness (right eye, INL_thickness_right, index variant rs7752421) and schizophrenia (index variant rs3798869). Bayesian colocalization analysis suggested the shared causal variant between the two traits (posterior probability PPH4 = 0.952). In this region, the thickness of INL was also in LD ($r^2 \geq 0.6$) with bipolar disorder and cognitive ability. **B** In 14q11.2, shared genetic influences were observed between the overall macular thickness (right eye, overall_macular_thickness_right, index variant rs200581586) and educational attainment (index variant rs4982712, PPH4 = 0.764). In this region, the overall macular thickness was also in LD ($r^2 \geq 0.6$) with intelligence and cognitive

ability. **C** In 9p21.3, shared genetic influences were observed between the vertical cup-to-disc ratio (regressed on disc diameter, left eye, VCDR_regressed_left, index variant 9:22053956_TA_T) and Glioma (index variant rs4977756). We also observed genetic overlaps (LD $r^2 \geq 0.6$) with self-reported math ability, risk-taking tendency, and loneliness. **D** In 13q14.13, shared genetic influences were observed between the fifth PC of the Vgg16 model on fundus image (Vgg16_PC5, index variant rs866376) and ischemic stroke (index variant rs9526212, PPH4 = 0.994). The dotted lines represent different significance levels in GWAS studies of these phenotypes, where the genetic effects were assessed using either a linear model or a linear mixed-effects model in a two-sided test.

cognitive impairment and long-term cognitive decline in older individuals[88,89]. Retinal imaging traits were in LD ($r^2 \geq 0.6$) with multiple neurodegenerative disorders, such as in 2q24.3, 17q21.31, 8p23.1, and 15q12 with Parkinson's disease[90] (Fig. S76); in 17q21.31 with corticobasal degeneration[91]; in 7p21.3 with frontotemporal dementia[92]; in 5q14.3, 17q21.31, 6p12.1 with Alzheimer's disease[93], and several more loci (such as 1q32.1, 2p25.3, and 11q14.2) with biomarkers of Alzheimer's disease[94] (Figs. S77–79). Genetic overlaps with other brain diseases were also observed. For example, the vertical cup-to-disc ratio[30] was in LD ($r^2 \geq 0.6$) with glioma/ glioblastoma[95] and white matter microstructure in 9p21.3 (Fig. 6C). Glioma can affect the optic nerve (optic nerve glioma), which is the most common primary neoplasm of the optic nerve[96]. Retinal imaging traits also shared genetic effects with migraine/headache[97] in 5 regions and cerebrovascular diseases in 9 regions, including stroke[98], Moyamoya disease[99], intracranial aneurysm[100], and cerebral aneurysm[101] (Figs. 6D and S80–82).

## Genetic correlation and heritability enrichment patterns

We examined genetic correlations (GC) between 156 retinal imaging traits and 39 sets of publicly available GWAS summary statistics of brain-related complex traits and diseases using cross-trait LDSC[102] (Supplementary Data 9). At a 5% FDR level ($P < 2.06 \times 10^{-3}$), we observed 246 significant genetic correlation pairs between 69 retinal imaging traits and 21 brain phenotypes, including brain disorders, cognitive traits, and mental health traits (Fig. S83).

Multiple cognitive traits (such as cognitive function, numerical reasoning, intelligence, and cognitive performance) and education consistently exhibited positive genetic correlations with the thickness of RNFL[44] and the overall thickness between the ELM to the inner and outer photoreceptor segments (ISOS)[44], as well as their subfields. Consistent with our results, previous clinical studies have identified RNFL thickness to be phenotypically related to global cognitive score, executive function, and verbal function[103–105]. These studies examined RNFL thickness as a possible early biomarker of cognitive decline, whose thinning suggests axonal loss during the neurodegenerative process of the brain[105,106]. On the other hand, negative genetic correlations with cognitive traits were observed for the thickness of GCIPL, INL, and RPE[44], as well as disc diameter[30]. The negative correlations between GCIPL thickness and cognitive traits were also in line with one recent study on patients with Alzheimer's disease, where GCIPL thickening correlated with poor cognition in Alzheimer's disease[107]. One hypothesis on the intrinsic mechanism for its thickening suggested that pathological Aβ accumulation and neuroinflammation of retinal ganglion cells (RGCs) contributed to the thickening of GCIPL[107], which was supported by a parallel study on RGCs in mouse model[108].

The thickness between the ISOS and RPE and their subfields were negatively associated with depression and neuroticism. Additionally, negative associations were found between depression symptoms and vertical cup-to-disc ratio[30] (GC < −0.166, $P < 7.72 \times 10^{-4}$), between cross disorder (five major psychiatric disorders[109]) and disc diameter (GC = −0.116, $P = 1.33 \times 10^{-3}$), and between the RNFL thickness and cannabis use disorder (GC = −0.174, $P = 1.29 \times 10^{-3}$). In addition, positive genetic correlations were discovered between RPE thickness and attention-deficit/hyperactivity disorder (ADHD), as well as between the INL thickness and stroke (GC > 0.149, $P < 1.88 \times 10^{-3}$). For fundus imaging traits, widespread genetic correlations with the aforementioned brain phenotypes identified by OCT measures were also observed, such as ADHD, cannabis use disorder, cognitive traits, and cross disorder. In addition, fundus imaging traits had higher correlations with schizophrenia (|GC| = 0.133, $P = 1.10 \times 10^{-3}$), major depressive disorder (|GC| = 0.316, $P = 1.92 \times 10^{-3}$), and risk tolerance (|GC| = 0.097, $P = 1.92 \times 10^{-3}$). These results demonstrated the genome-wide genetic similarity between retinal structures and brain disorders and traits. The observed genetic correlations between retinal imaging traits and various brain-related complex traits and diseases further highlight the

potential of retinal assessments as valuable biomarkers for these conditions.

We also performed a partitioned heritability analysis[110] using LDSC to determine the tissues and cell types, in which genetic variation led to changes in retinal imaging traits. First, we examined a wide variety of tissue and cell type-specific regulatory elements from the Roadmap Epigenomics Consortium[111]. Among all tissue and cell types, the strongest heritability enrichments were observed in active gene regulatory regions of multiple brain tissues (Fig. S84 and Supplementary Data 10). Next, we repeated the partitioned heritability analysis using chromatin accessibility data from neurons (NeuN+) and glia (NeuN−) sampled from 14 cortical and subcortical brain regions[112]. We observed that the heritability of retinal structures showed consistently stronger enrichment in brain glial regulatory elements than neuronal regulatory elements (Fig. S85). These heritability enrichments suggest that genetic variants associated with retinal structures may also alter the function of regulatory elements in brain tissues, particularly glial cells.

## Genetic causal links with brain disorders

We applied Mendelian randomization (MR) with GWAS summary statistics from the FinnGen database[113] to examine the directional relationships between retinal structure and brain disorders. We used eight different MR methods[114–121], and prioritized significant results that passed the Bonferroni adjustment of multiple testing in at least two methods ("Methods" section). The results presented below have also passed several robustness tests, such as the MR-Egger intercept test for pleiotropy[122].

Causal genetic effects were identified between retinal imaging traits and brain disorders in both directions, suggesting close relationships between retinal structures and Alzheimer's disease (Supplementary Data 11). For example, causal genetic effects from Alzheimer's disease to retinal structures were found in various OCT measures and fundus imaging traits, including the thickness of INL ($\beta > 0.025$, $P < 4.74 \times 10^{-5}$) and the central subfield between the ISOS and RPE ($\beta > 0.027$, $P < 1.12 \times 10^{-5}$). We also observed causal effects from psychiatric diseases and other degenerative diseases of the nervous system to retinal structures, such as the INL thickness ($\beta > 0.040$, $P < 3.56 \times 10^{-7}$). These newly established positive causal effects between psychiatric diseases and INL thickness can be linked to the identified negative genetic correlations between INL thickness and cognitive traits in our previous section. There was a similar conclusion reached in previous studies regarding the thickness of the RNFL, whose thinning was indicative of cognitive decline[123,124]. On the contrary, other recent studies have also noted correlations between INL thickening and brain-related diseases, such as Alzheimer's disease and multiple sclerosis[125,126]. These studies suggested that INL thickness was a response marker for inflammation during the early stages of diseases, which was further confirmed by another study, where effective disease treatment was associated with a reduction in INL thickness[127]. In addition, when using retinal imaging traits as exposures and brain disorders as outcomes, we found causal effects from retinal structural changes to dementia and Alzheimer's disease. These causal links were observed on fundus imaging traits generated from pre-trained transfer learning models. For OCT measures, causal links were identified between anxiety disorders and the thickness of the central subfield between the INL and RPE ($\beta = 0.278$, $P = 7.92 \times 10^{-6}$). Overall, MR analysis indicates that retinal imaging traits have genetic interactions with neurodegenerative and neuropsychiatric diseases, especially dementia and Alzheimer's disease.

## Joint prediction of brain phenotypes using retinal and brain imaging

By utilizing both retinal and brain imaging traits, we investigated whether combining these data types could lead to better predictions of brain-related complex traits and diseases compared to using just

only one type of imaging data. We employed a training, validation, and testing design, in which both retinal and brain images were available for subjects in the validation and testing datasets. Model parameters were fine-tuned based on the validation data, and prediction performance was assessed in the independent testing dataset ("Methods" section).

First, retinal imaging traits demonstrated significant predictive power for 16 brain-related phenotypes, including cognitive traits (such as fluid intelligence and prospective memory), neuroticism, family history of stroke, mental and behavioral disorders (ICD-10 Chapter F, such as depressive episode), and diseases of the nervous system (ICD-10 Chapter G, such as multiple sclerosis and carpal tunnel syndrome) (prediction correlation $\beta$ range = [0.068, 0.179], $P$ = [8.11 × 10$^{-19}$, 7.88 × 10$^{-4}$], Fig. S86 and Supplementary Data 12). The strongest prediction accuracy was observed for fluid intelligence ($\beta$ = 0.179, $P$ = 8.11 × 10$^{-19}$). The top-ranking features for fluid intelligence prediction were derived from both OCT measures and fundus imaging traits, such as the thickness of RNFL, INL, and GCIPL. Moreover, the prediction accuracy was improved by adding more retinal imaging traits, suggesting that various retinal structural variations captured by different retinal imaging modalities and pre-trained models can contribute to cognitive performance prediction (Fig. S87A). Similar additive effects were observed for other brain phenotypes, such as the family history of stroke (Fig. S87B). Multiple clinical studies have suggested that retinal imaging traits (such as retinal layer thickness) show promising prediction power for pathological cognitive decline and dementia diagnoses[8,106].

Next, we included brain MRI traits in the prediction model of these brain-related phenotypes. Figure S88A shows that multimodal brain imaging data can significantly predict all these brain phenotypes ($\beta$ range = [0.091, 0.314], $P$ range = [7.61 × 10$^{-6}$, 3.25 × 10$^{-56}$]), and using both retinal and brain imaging traits can further improve the performance ($\beta$ range = [0.120, 0.344], $P$ range = [3.72 × 10$^{-9}$, 7.54 × 10$^{-68}$]). For example, multiple categories of brain MRI traits can predict fluid intelligence, including DTI parameters ($\beta$ = 0.118, $P$ = 7.05 × 10$^{-9}$), regional brain volumes ($\beta$ = 0.132, $P$ = 8.17 × 10$^{-11}$), cortical thickness traits ($\beta$ = 0.100, $P$ = 9.01 × 10$^{-7}$), resting fMRI ($\beta$ = 0.216, $P$ = 7.01 × 10$^{-27}$), and task fMRI ($\beta$ = 0.197, $P$ = 2.11 × 10$^{-22}$). Adding retinal imaging traits to each of these brain modalities improved the prediction performance over only using this single brain modality. The largest improvement was observed when we added all imaging data types together ($\beta$ = 0.344, $P$ = 7.54 × 10$^{-68}$). The prediction accuracy further moved up to 0.391 ($P$ = 9.56 × 10$^{-89}$) by adding the genetic PRS of fluid intelligence (Fig. S88B). These results demonstrate that integrating retinal and brain imaging modalities may lead to better predictions of brain-related complex traits and diseases than using only one type of imaging data alone.

## Discussion

Imaging of the eye is inexpensive and noninvasive, and it can provide rich information about the retina's structure and function. Many brain diseases, such as neuropsychiatric and neurodegenerative disorders, are diagnosed and monitored primarily based on subjective reports of clinical symptoms[128]. The accuracy of these subjective reports is often complicated by the fact that patients with impaired mental capacity report inconsistent symptoms in varying degrees, which can bias the downstream data analysis and clinical prediction[129]. Moreover, patients presenting with acute mental symptomatology may have hard-to-define underlying ailments, leading to imprecise medical management. Retinal imaging traits may serve as objective biomarkers for brain abnormalities and to assess the progression of neurological conditions[130].

In this paper, we analyzed eye-brain connections using multimodal imaging data from both organs. The pericalcarine (primary visual cortex) and other structures within the visual pathway were

associated with retinal features. Furthermore, we observed correlations between retinal features and thalamic volume, both of which are derived from the diencephalon. We then described the genetic co-architecture of the eye and the brain in 65 genomic regions, suggesting genetic associations that overlap among retinal features, brain MRI traits, and eye and brain disorders. We discovered genetic correlations and causal links between retinal imaging traits and various cognitive and mental health traits, as well as brain disorders. Additionally, we demonstrated that multi-organ images could be combined to improve the prediction of brain phenotypes. These analyses represent a significant step forward in understanding eye-brain interactions. Our results are consistent with previous studies that have found parallel changes in eye-brain structures during pathological progression[1,3], as well as providing further information on the most relevant imaging modalities and phenotypes for future research. We also reveal close genetic connections between the eye and the brain, and abnormalities in retinal structure may provide insight into the genetic risk of neurodegenerative diseases and neuropsychiatric disorders. Together, these understandings could lead to improved diagnostic and treatment strategies for brain conditions by leveraging retinal imaging traits as potential phenotypes.

Specifically, the discovery of retinal imaging phenotypes and their shared genetic risk factors with brain disorders could pave the way for less invasive monitoring and early detection methods, providing valuable insights into the pathological processes of these conditions. For example, we found that the thickness of various retinal layers, such as the RNFL, shared genetic influences with schizophrenia across multiple genomic loci (Figs. 6A and S40–S43). It is known that individuals with schizophrenia often exhibit thinner retinal layers and reduced macular thickness[76,131]. Beyond schizophrenia, these retinal layers also demonstrated extensive genetic associations with neuroticism (Figs. S44–S50). Such insights underscore the potential clinical utility of retinal imaging phenotypes and their genetic risk scores in the diagnosis and management of psychiatric disorders and mental health issues.

Additionally, our findings reveal close genetic links between retinal imaging traits and neurovegetative disorders, including Parkinson's disease and Alzheimer's disease. These results align with recent research suggesting retinal measures as promising tools for early assessment of Parkinson's disease pathogenesis[132,133] and corroborate existing cohort studies indicating that retinal neurodegeneration markers could assist in identifying individuals at elevated risk for Alzheimer's disease and related dementia[6,8,16–19,134]. Furthermore, we discovered genetic links to glioma/glioblastoma, a type of tumor known to predominantly spread along brain white matter, aligning with the observed strong phenotypic correlations between retinal imaging traits and DTI parameters. Retinal imaging could enhance the early detection and stratification of neurological conditions, leading to improved patient outcomes.

In addition to brain disorders, our analysis revealed associations between retinal imaging traits and various brain MRI, cognitive, and mental health traits, as well as showcasing predictive power over several domains. These findings underscore the potential of retinal imaging for monitoring brain health and well-being across a wide range of ages in the general population, extending its applications beyond the diagnosis of specific diseases. Moreover, the pronounced genetic basis of these links suggests the feasibility of applying PRS of retinal imaging in wider genetic cohorts lacking real imaging data[135]. Future studies can delve further into building more powerful retinal imaging-based prediction models for the brain and evaluating their performance in the growing body of data resources.

This study has a few limitations. Our analyses were based on the ongoing UKB brain imaging study, which currently covers only a small proportion of all UKB participants and consists primarily of individuals of European ancestry. We conducted phenotypic analyses on an even smaller sample of UKB subjects with both eye and brain imaging data.

It is anticipated that more brain-related retinal imaging biomarkers can be discovered and replicated as the UKB brain imaging study collects data from additional subjects[136]. Furthermore, inferring phenotypic causality between retinal and brain imaging phenotypes from our current cross-sectional analysis is challenging. Repeated UKB eye and brain imaging scans[137] in the future will enable us to study the causal relationships between eye and brain structural and functional changes using a longitudinal study design.

In addition, the two-sample MR method used in this study, an epidemiological approach for genetic causal inference, relies on assumptions on genetic variants and may not offer full insights into inter-organ biological processes. Insights at the molecular and cellular levels are needed to better understand the causal connections between the eye and the brain[138]. In addition, the eye-brain genetic links identified were specific to European or UKB populations, and it will be important to examine whether these cross-organ genetic overlaps can be generalized to other populations or studies as more data are collected[139]. In summary, our UKB-based study reveals substantial genetic connections between the brain and the eye, suggesting the potential clinical applications of retinal imaging not only in neuropsychiatric and neurological practice but also in routine brain health care for the general population. The utility of these retinal imaging phenotypes needs to be verified in future longitudinal studies and clinical settings.

## Methods
### Eye and brain imaging data
Our study was based on data obtained from the UK Biobank (UKB) study, which recruited approximately half a million individuals between the ages of 40 and 69 between 2006 and 2010[43] (https://www.ukbiobank.ac.uk/). The ethics approval of the UKB study was from the North West Multicentre Research Ethics Committee (approval number: 11/NW/0382) and informed consent was obtained by participants. The optical coherence tomography (OCT) and retinal imaging scans were part of the eye measurements conducted during the participant's visit to the UKB assessment center. We considered two sets of retinal imaging traits. First, we used the derived OCT measures in Category 100079, which were generated and returned by previous studies[30,44,45]. These measures mainly provide the thickness of different retinal layers and their subfields, as well as the vertical cup-to-disc ratio and disc diameter. As suggested, we used the data in Data-Fields 28552 & 28553[44] to perform quality control for these OCT measures by keeping images with an image quality score >45. We further only keep the OCT measures with a sample size >30,000, resulting in 46 measures with an average sample size of 62,425.

Second, we downloaded the raw fundus retinal eye images from Category 100016 and performed GWAS on these whole images by extracting imaging biomarkers using transfer learning models. Briefly, we used multiple pre-trained deep convolutional neural networks (CNNs) trained from the ImageNet[46] database. The ImageNet database contains more than 14 million images classified into more than 20,000 classes, which can be used to train models that extract various features from retinal fundus images. Many CNNs models have been trained on ImageNet and were widely used in the image processing field to learn complex patterns from images. In addition to the ResNet50[140] model used by the transferGWAS[32], we implemented 10 more pre-trained CNN models, including the AlexNet[141], Vgg16[142], Vgg19[142], GoogLeNet (Inception V1)[143], Inception (V3)[143], ResNet18[140], ResNet34[140], SqueezeNet[144], MobileNet[145], and ShuffleNet[146]. These pre-trained models are available on Pytorch[147] and represent different designs and architectures, such as layer depth, size of kernels, and hyperparameters. For example, the ResNet50 has 50 layers with kernel size $1 \times 1$, $3 \times 3$, and $7 \times 7$, while AlexNet has 8 layers with kernel size $3 \times 3$, $5 \times 5$, and $11 \times 11$. All pre-trained models use the rectified linear unit (ReLU) as the activation function. We began by combining the original

left and right retinal fundus images and the rotated images with 90°, 180°, and 270°, each with and without horizontal mirroring. Next, we input these eight retinal fundus images into each pre-trained model and averaged the outputs from the last layer of convolutional networks. Then we generated the top-10 ranked principal components (PCs) from each of the 11 models as retinal imaging biomarkers in downstream GWAS analyses. The average sample size across all these 110 ($10 \times 11$) fundus imaging traits is 78,513. In all the OCT measures and fundus image traits, the values greater than five times the median absolute deviation from the median were treated as outliers and removed.

The UKB brain MRI data were generated from raw images downloaded from Category 100003. The multimodal brain imaging traits used in the present paper have been extracted in previous papers by our research group[36,38,40,47]. First, we had 101 regional brain volumes[36] and 63 cortical thickness traits[47] generated from T1-weighted structural MRI images. These structural MRI traits were produced by the ANTs[48]. For the 101 volumetric traits, we had brain volumes for 98 predefined cortical and subcortical areas and three global brain volume measures (total gray matter volume, total white matter volume, and total brain volume). We also examined the thickness of 62 cortical areas and the global thickness. Second, the ENIGMA-DTI pipeline[148,149] was used to generate 110 tract-averaged DTI parameters based on diffusion MRI, including fractional anisotropy, mean diffusivity, axial diffusivity, radial diffusivity, and mode of anisotropy, for 21 predefined major white matter tracts and the whole brain ($5 \times 22$). For resting fMRI, we applied the Glasser360 atlas[150] to partition the cerebral cortex into 360 regions for 12 functional networks[151], including the primary visual, secondary visual, auditory, somatomotor, cinguloopercular, default mode, dorsal attention, frontoparietal, language, posterior multimodal, ventral multimodal, and orbito-affective networks. We generated 92 functional activity (amplitude) and functional connectivity traits, including the average activity for each network and the average connectivity for each pair of networks (including within the same network), as well as the global activity and connectivity of the whole cortex. Similarly, 92 functional activity and connectivity traits were generated from task fMRI[40]. In summary, we considered 458 brain MRI traits of brain structure and function. See Supplementary Data 1 for the complete ID list of both retinal imaging and brain imaging traits.

### Phenotypic eye-brain imaging analyses
In our phenotypic analysis, we examined pairwise associations between 156 retinal imaging traits and 458 brain MRI traits. We used the UKB subjects with both two imaging types and adjusted a wide range of covariates, including age, sex (self-reported), standing height, assessment center, body mass index, weight, waist-to-hip ratio, smoking status, mean arterial blood pressure, age-squared, age-sex interaction, age-squared-sex interaction, top 40 genetic PCs[152], volumetric scaling, head motion, head motion-squared, brain position, brain position-squared[35,37], diabetes, ICD-10 disease code staring with R73 ("elevated blood glucose level", such as hyperglycemia), I70 ("atherosclerosis", such as atherosclerosis of aorta), I10 (Essential (primary) hypertension), and E78 ("disorders of lipoprotein metabolism and other lipidaemias", such as hyperlipidemia). For regional brain volumes, we additionally corrected for total brain volume to remove global effects.

We fitted linear models for each pair of imaging traits (R version 3.6.0) and used a discovery-validation design, in which the UKB individuals of white British ancestry (average $n = 6454$ across different modalities) were used to discover eye-brain imaging associations, which were verified by a hold-out independent validation dataset (average $n = 959$). This hold-out dataset included all subjects not used in the discovery dataset, encompassing various ancestry backgrounds. Additionally, relatives[152] of individuals from the discovery sample were

excluded. The Benjamini–Hochberg procedure was used to adjust for multiple testing, and we reported significant associations at the false discovery rate (FDR) of 5%. Validation criteria included a *P*-value less than 0.05 in the hold-out independent dataset with concordant association signs between the discovery and validation datasets. We also considered the conservative Bonferroni multiple testing correction and highlighted these top-ranking significant findings in the paper. In addition, we repeated the above analysis separately for females and males (average *n* = 3338 and 3150, respectively) and reported the sex-specific association patterns. We also performed additional sensitivity analysis to add ocular and brain disorders as covariates (Supplementary Note).

## Genetic analysis of 156 retinal imaging traits

We performed GWAS for the 156 retinal imaging traits using the imputed genotyping data from the UKB study. For the set of subjects with both retinal imaging traits and genetic data, we performed the following quality controls[40]: (1) removed individuals with missing genotype rate >0.1; (2) removed variants with missing genotype rate >0.1; (3) removed variants with minor allele frequency (MAF) < 0.01; (4) removed variants that failed the Hardy-Weinberg equilibrium test at $1 \times 10^{-7}$ level; and (5) removed variants with imputation INFO score <0.8. The SNP-based heritability of white British samples was estimated based on all autosomal SNPs using GCTA[53] (average *n* = 60,748). We adjusted for the effects of age, sex, assessment center, age-squared, age-sex interaction, age-squared-sex interaction, and top 40 genetic PCs[152].

Using the same set of subjects and covariates data, we performed GWAS using linear mixed effect models via fastGWA[153]. SNP heritability and GWAS were also conducted separately for males and females. We defined the independent (LD $r^2 < 0.1$) significant genetic associations and loci using FUMA[154] (version v1.3.8). The details of FUMA annotations can be found at https://fuma.ctglab.nl/tutorial. Briefly, FUMA first labeled independent significant variants, which were variants whose *P*-values were smaller than $3.20 \times 10^{-10}$ and were independent of other significant variants (LD $r^2 < 0.6$). Next, LD boundaries were defined by tagging all variants (MAF ≥ 0.0005, including those from the 1000 Genomes reference panel) in LD ($r^2 \geq 0.6$) with at least one of these independent significant variants. Based on the independent significant variants, FUMA further defined independent lead variants as those that were independent of others (LD $r^2 < 0.1$). If there were independent significant variants that were close to each other (<250 kb based on their LD boundaries), FUMA merged their LD blocks into one single genomic locus to obtain the LD boundaries. For independently significant associations and LD blocks defined by FUMA, we performed validations using (1) the UKB European but non-British subjects (average *n* = 5320) and (2) UKB non-European subjects from different ancestry backgrounds (average *n* = 6490). The ancestry information was derived from self-reported ethnicity (Data-Field 21000), the accuracy of which was confirmed in Bycroft et al. [152]. Relatives of the discovery GWAS sample were removed, and we adjusted for the same set of covariates as in the discovery sample, including the top 40 genetic PCs to account for population stratifications. We also developed PRS using summary statistics from the discovery of GWAS and examined their prediction accuracy on the two validation datasets. We constructed polygenic risk scores (PRS) based on PRS-CS[56] with all default parameters. To avoid overlapping with the training discovery GWAS sample, the validation genotype data were randomly selected from 1500 UKB European subjects without retinal imaging data.

## Genetic eye-brain imaging analyses

For the independently significant variants and all variants in their LD blocks, we used FUMA to look them up on the NHGRI-EBI GWAS catalog (version e104_2021-09-15) to search for any previous GWAS

results reported on these variants ($P < 9 \times 10^{-6}$). We focused on the existing GWAS results of brain and eye-related complex traits and diseases and manually categorized them into 14 groups, including stroke (and other cerebrovascular disorders, such as Moyamoya disease, intracranial aneurysm, and cerebral aneurysm), Parkinson's disease, Alzheimer's disease, glioma/glioblastoma (GBM), other neurological disorders (such as amyotrophic lateral sclerosis, progressive supranuclear palsy, corticobasal degeneration, and frontotemporal dementia), schizophrenia, other psychiatric disorders (such as bipolar disorder, depression, major depressive disorder, and autism spectrum disorder), psychological traits (such as neuroticism, anxiety, subjective well-being, and risk tolerance), cognitive traits (such as general cognitive ability, the highest math class taken, intelligence, and reaction time), education, brain structure/function, migraine, Alzheimer's disease biomarkers (such as cerebrospinal fluid biomarker levels, rate of cognitive decline in Alzheimer's disease, and plasma t-tau levels), and eye traits/diseases (such as macular thickness, refractive error, spherical equivalent, and glaucoma). In addition, we systematically examined genetic overlaps with the GWAS results of brain MRI traits reported in previous studies, including 101 regional brain volumes[36], 215 DTI parameters[38] (including the 110 tract-average values used in our phenotypic analysis and 105 additional PCs of fractional anisotropy), 63 cortical thickness traits[47], 92 resting fMRI traits, and 92 task fMRI traits[40]. For the index variants of retinal imaging traits defined by FUMA, we looked up the MetaBrain database[64] (https://www.metabrain.nl/) to see if they were reported eQTLs in large-scale gene expression meta-analysis of brain tissues. For each locus with shared genetic influences, we tested for common causal genetic variants between the retinal imaging trait and the brain phenotype using Bayesian colocalization analysis[62]. The colocalization was established if the posterior probability of the shared causal variant hypothesis (PPH4) was greater than 0.8[62,63].

Cross-trait LDSC[102] (https://github.com/bulik/ldsc/, version 1.0.1) was used to examine the pairwise genetic correlation between 156 retinal imaging traits and 39 sets of publicly available GWAS summary statistics of brain phenotypes. The default European LD scores provided by the LDSC software were used, which were based on the HapMap3 genetic variants and were estimated from the 1000 Genomes European samples. The major histocompatibility complex region was excluded due to the complex LD pattern. For the 46 OCT measures, we also used LDSC to perform the heritability enrichment analysis[110] with genetic variant annotations of tissue type and cell type-specific regulatory elements. The heritability explained by the annotated genome regions was estimated and tested with percentages and enrichment scores. Baseline annotation models were included in the analysis when we analyzed additional annotations. We tested for the annotations of regulatory elements from multiple adult and fetal tissues from the Roadmap Epigenomics Consortium[111] and two major brain cell types (neurons and glia) sampled from various brain cortical and subcortical brain regions[112].

Bi-directional Mendelian randomization (MR) analysis was used to discover the causal effect between 156 retinal imaging traits and 25 brain-related clinical endpoints. Eight MR methods[114–121] were implemented, including MR Egger, simple median, simple mode, fixed effect inverse variance weighted (IVW), multiplicative random effect IVW, DIVW, MR-RAPS, and GRAPPLE. The 25 brain-related clinical endpoints were all from the latest release (R7) of the FinnGen database (https://www.finngen.fi/en/access_results), where 12 of them were mental and behavior disorders, and the remaining 13 phenotypes were diseases of the nervous system. Most of the diseases we selected have a number of cases greater than 10,000, except for a few important brain diseases, including Alzheimer's disease (*n* > 6000), other neurological diseases (*n* = 7288), and epilepsy (*n* = 8523). Supplementary Data 11 provides more information on the MR methods and FinnGen data. Exposure GWAS summary statistics were first clumped with Plink[155] to guarantee

that the instrumental variables used in MR models are independent. The P-value significance threshold (p1) and the secondary significance threshold (p2) in clumping were set to $5 \times 10^{-8}$, and the 1000 Genomes European reference panel was applied. Besides, the threshold over the squared correlation between two genetic variants was set to be $r^2 = 0.01$ and window size = 1 Mb. After clumping, the selected SNPs from exposure GWAS data were extracted from outcome GWAS summary statistics with function *extract_outcome_data()* in the two-sample MR package (https://mrcieu.github.io/TwoSampleMR/). To ensure that the effect of a genetic variant on the exposure and outcome corresponded to the same allele, data harmonization was performed using the *harmonise_data function()* with the default settings. The estimated causal pairs of retinal imaging trait and brain disease were further screened with several rules. The first step was to discard pairs with fewer than six genetic variants. Second, we dropped the pairs whose estimated MR Egger intercept differed significantly from zero[122]. Bonferroni correction was then performed on the MR results of each method separately. Finally, we reported the causal pairs that were significant for either of the IVW methods and at least one of the robust MR methods (DIVW, simple mode, simple median, MR-RAPS, and GRAPPLE).

### Prediction of brain phenotypes using retinal and brain imaging data

We examined the prediction power of 156 retinal imaging traits on 32 brain-related complex traits and diseases, including cognitive traits, neuroticism sum score, family history of brain disorders, mental and behavioral disorders (ICD-10 Chapter F), and diseases of the nervous system (ICD-10 Chapter G). We focused on unrelated white British subjects and first randomly divided the subjects who had both retinal and brain imaging data into validation and testing datasets (2464 subjects in each dataset). This design enabled us to test the prediction performance using both imaging types in later steps. The remaining 50,944 subjects with retinal imaging data (but no brain MRI data) were used as the training dataset. For each of the 32 traits, we used ridge regression for prediction, and the effect sizes of retinal imaging traits were estimated on the training dataset via the glmnet[156] package (R version 3.6.0). All model parameters were tuned based on validation data, and prediction performance was examined based on the correlation between the predicted values and the observed ones in the independent testing data. In all the training, validation, and testing datasets, we removed the effects of age, sex, age-sex interaction, age-squared, age-squared-sex interaction, assessment center, and top 40 genetic PCs. For brain phenotypes where retinal imaging traits had significant predictive power after Bonferroni correction for multiple testing, we further examined the predictive power of multiple brain MRI modalities and the joint performance of using retinal and brain imaging traits. In brain MRI prediction, we used the same validation and testing datasets as the above retinal imaging analysis (2464 subjects in each dataset) and all other unrelated white British subjects (37,239 subjects) as training data. Finally, we examined the prediction accuracy of genetic PRS for fluid intelligence. We used the unrelated white British subjects without retinal or brain imaging data as training GWAS (n = 71,406) and developed the PRS with PRS-CS[56]. The same set of covariates as the imaging prediction analysis was removed.

### Reporting summary

Further information on research design is available in the Nature Portfolio Reporting Summary linked to this article.

## Data availability

The GWAS summary statistics of retinal imaging traits generated in this study have been deposited in the Zenodo database under accession code 11217687. The GWAS summary statistics of brain MRI traits can be freely downloaded at BIG-KP (https://www.med.unc.edu/bigs2/data/ gwas-summary-statistics/). The individual-level UK Biobank imaging data used in this study can be obtained from https://www.ukbiobank.ac.uk/.

## Code availability

We made use of publicly available software and tools. The codes to apply pre-trained transfer learning models to extract features from raw retinal fundus images are available at https://github.com/mkirchler/transferGWAS.

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

## Acknowledgements

We thank Mufeng Gao for her help with data management in the early stages of this project. Figures 1 and 3B were created with BioRender.com. Matplotlib[157] contributed to Fig. 2. Research reported in this publication was supported by the National Institute On Aging of the National Institutes of Health under Award Number RF1AG082938 (B.Z. and H.Z.). The content is solely the responsibility of the authors and does not necessarily represent the official views of the National Institutes of Health. The study has also been partially supported by funding from the Purdue University Statistics Department, Department of Statistics and Data Science at the University of Pennsylvania, and Analytics at Wharton (B.Z.). The study has also been partially supported by the National Institute On Aging of the National Institutes of Health under Award Numbers U01AG079847 and R01AR082684 (Yun.L. and H.Z.). Assistance for this project was also provided by the UNC Intellectual and Developmental Disabilities Research Center (NICHD; P50 HD103573; Yun.L.). This research has been conducted using the UK Biobank resource (application number 22783), subject to a data transfer agreement. We would like to thank the individuals who represented themselves in the UK Biobank for their participation and the research teams for their efforts in collecting, processing, and disseminating these datasets. We would like to thank the research computing groups at the University of North Carolina at Chapel Hill, Purdue University, and the Wharton School of the University of Pennsylvania for providing computational resources and support that have contributed to these research results. We gratefully acknowledge all the studies and databases that made GWAS summary-level data publicly available.

## Author contributions

B.Z. designed the study. Yujue.L. and B.Z. processed the raw retinal imaging data. B.Z., Z.F., Z.W., Juan.S., X.Y., Xifeng.W., B.L., Xiyao.W., and C.C. analyzed the data. Z.F., B.Z., Yue.Y., and J.L. designed the website and developed online resources. Yilin.Y., Yun.L., Jason.S., J.M.O., T.L., and H.Z. provide comments and interpret the results. B.Z. wrote the manuscript with feedback from all authors.

## Competing interests

The authors declare no competing interests.

## Additional information

[1]Department of Statistics and Data Science, University of Pennsylvania, Philadelphia, PA 19104, USA. [2]Department of Statistics, Purdue University, West Lafayette, IN 47907, USA. [3]Applied Mathematics and Computational Science Graduate Group, University of Pennsylvania, Philadelphia, PA 19104, USA. [4]Center for AI and Data Science for Integrated Diagnostics, Perelman School of Medicine, University of Pennsylvania, Philadelphia, PA 19104, USA. [5]Penn Institute for Biomedical Informatics, Perelman School of Medicine, University of Pennsylvania, Philadelphia, PA 19104, USA. [6]Population Aging Research Center, University of Pennsylvania, Philadelphia, PA 19104, USA. [7]Institute for Translational Medicine and Therapeutics, University of Pennsylvania, Philadelphia, PA 19104, USA. [8]Department of Biostatistics, University of North Carolina at Chapel Hill, Chapel Hill, NC 27599, USA. [9]Department of Computer Science, Purdue University, West Lafayette, IN 47907, USA. [10]Yale School of Management, Yale University, New Haven, CT 06511, USA. [11]Department of Genetics, University of North Carolina at Chapel Hill, Chapel Hill, NC 27599, USA. [12]UNC Neuroscience Center, University of North Carolina at Chapel Hill, Chapel Hill, NC, USA. [13]Scheie Eye Institute, University of Pennsylvania, Philadelphia, PA 19104, USA. [14]Penn Medicine Center for Ophthalmic Genetics in

Complex Diseases, Philadelphia, PA 19104, USA. [15]Department of Radiology, University of North Carolina at Chapel Hill, Chapel Hill, NC 27599, USA. [16]Biomedical Research Imaging Center, School of Medicine, University of North Carolina at Chapel Hill, Chapel Hill, NC 27599, USA. [17]Department of Computer Science, University of North Carolina at Chapel Hill, Chapel Hill, NC 27599, USA. [18]Department of Statistics and Operations Research, University of North Carolina at Chapel Hill, Chapel Hill, NC 27599, USA. ✉e-mail: bxzhao@wharton.upenn.edu; htzhu@email.unc.edu

