## [Peer Review File · Nature Communications]

Eye-brain connections revealed by multimodal retinal and brain imaging geneticsREVIEWER COMMENTS

Reviewer #1 (Remarks to the Author):

Zhao et al has looked into retinal imaging biomarkers and compared to the brain structure and function modalities. They examined images from the UK biobank (fundus and OCT) (n=6,454 NHW individuals) with a full GWAS for those traits, and then correlated those with MRI modalities from brain. Associations were found for 121 tests that 66 replicated in an independent dataset. This information shows the clear relationship between eye and brain phenotypes, as well as the association between eye imaging features and brain, both of which can be used to diagnose disorders. Heritability was also determined for the eye features found in OCT. The independent datasets were in non-European multi ancestry datasets (n=6,490) as well as those of European ancestry who were not British (n=5320). There is a huge association of different genetics with both brain and eye diseases, and the overlap in subsequent features that can then correlate with multiple diseases or phenotypes. This study is extremely interesting and shows a large overlap between those phenotypes, imaging modalities, and genetics that has not been previously shown. This work is of significance and novel, with fascinating methodology.

Some minor questions for clarity:

- 1) How genetically did you determine the European ancestry? PCs were adjusted for but is there a category in the UKB that indicates those specific ancestries? (Page 35 details)
- 2) The UKB participants have disease phenotypes. If those were not excluded, then of course different eye features or brain features would be found to correlate with disease genetics or with each other since those phenotypes have multiple features known to be included. How did you exclude/account for those? Some phenotypes were included (Page lines 11-16) and accounted for, but not for any ocular or brain disorder?
- 3) How did you handle the multi-ancestry in the UKB for the independent dataset? Was sub ancestry analysis performed?
- 4) It seems to be obvious that if you included those with retinal disease, the ocular features would indicate glaucoma (for example) and be correlated significantly (lines 20-27 etc page 9) – so were those samples separated out based on disease phenotype?
- 5) What is the hold-out independent dataset (n=959) – were these from both ancestry types? How were these determined? (Page 34 line 20)
- 6) Why were top 10 PCs used for the non European subjects instead of the 40? Due to ancestry? (Page 35 line 23-24)
- 7) Why was validation performed on the UKB European subjects without retinal imaging data? Did they have MRI data? (Page 35 line 28-29)
- 8) Why was HapMap 3 instead of 1000 genomes variants used for the LD scores? (Page 37 line 2)
- 9) How were the samples separated for training, validation and testing (Page 38 lines 14-16) from the entire UKB dataset?
- 10) Could the sex specific analysis have been weighted by disease that are found more predominantly in females than in males (for example the fundus image traits were the ones that had found extra associations)? (Page 6 lines 25-27).
- 11) How were the 142 regions determined (Page line 21-23) and the 140 regions? Are they the same regions?

Reviewer #2 (Remarks to the Author):

The authors aim to make eye-brain-genetic connections using the UKB to determine which structures and genes are related and how they might inform brain disorders. While the study is well designed, the execution of the results and the discussion are a major limitation. It may be worth describing some of the more significant findings and saving space to discuss them, as compared to forgoing on a

proper discussion of these very important results. For example, some very nice results relating to schizophrenia came up, but they are not discussed at all.

-The sentence should be reformatted "The retina, the only part of the central nervous system that can be visualized without surgical intervention, plays a critical role in the visual pathway." since it is not true that the brain cannot be visualized. This is done with MRI imaging for example. So it isn't clear why "surgical intervention" is mentioned. It is the case that retinal imaging is less invasive and provides greater detail than what is possible with brain MRI.

-would consider deleting the word altered in line 8 "As a result, the retina serves as a unique window into altered brain

9 structure/function1,3....."

-Were only healthy-ish people used for the analysis? If yes or no, please update line 12 where it states "We explored the genetic relationship between the eye and brain by analyzing multimodal retinal and brain imaging traits from the UK Biobank (UKB) study43."

-what retinal thickness measures were included? This would be important to note given their relationships between retinal layers and layers of the brain.

-Figure 3, some of the labels for A should be updated to reflect the correct nomenclature for the retinal layers. For example, what is ELM_ISOS_thickness?

-Figure 2, not sure why B and C start with describing DTI measures when in A, the analysis starts with volume. Please try to make the ordering consistent. I would make D, E, and F (volume and thickness findings) go first and then follow that up by B and C (DTI findings)

-why weren't Visual cortical regions extracted from the brain MRI measures. There are 2 tools built into FreeSurfer that can do this and it would have provided more accurate information related to V1 through V5/MT, as well as fusiform gyrus sub-regions, which have provided a more detailed map of these associations.

-fix "excitatory"

-the following sentence should be fixed for great clarity "As

22 neuroprotective treatments become more widely available, this ability to predict brain

23 diseases could have major clinical benefits."

-too much time is spent describing the results and very little time spent discussing what they might mean. there are much greater implications than what is currently discussed.

-the following sentence needs further clarifying so that it makes sense to readers who are not familiar with UKB data "Furthermore, inferring phenotypic causality from our current

11 cross-sectional analysis is challenging." In particular I am concerned about the inferences being made regarding diagnosis or other traits. For example, in the UK biobank, a diagnosis could have been identified either before or after imaging was performed.

Reviewer #3 (Remarks to the Author):

Zhao et al. describe a set of analyses, mainly based on the UK Biobank sample, examining the overlap between the genetic contributions to eye and brain structure. Specifically, they examined correlations between genetic findings from 156 retinal imaging variables (derived from color fundus photography and OCT), and 458 brain imaging variables (derived from structural MRI, DTI, and fMRI (resting state and task-based)). This is a timely and important paper, given the accumulating evidence that changes in the eye, especially in the retina, can concurrently or longitudinally predict changes in brain structure, brain function, and disease status (e.g., from no disease to a range of cardiovascular and neurodegenerative diseases; or from milder to more severe disease). The findings from this paper confirm the genetic bases for these effects. In addition, they suggest specific brain changes that are linked to retinal changes, and the genes involved in those relationships. Some of these findings are novel, such as the relationships between retinal findings and white matter tract integrity. The paper should stimulate further research on eye-brain connections, eye-brain disease connections, and the genetics of these connections. The findings are interesting, and the paper overall is a groundbreaking report.

This paper has many strengths. It is well-written and easy to follow. Prior literature is comprehensively cited, methods and results are described in detail, and the discussion sticks closely to the findings. Results are validated outside of the model training sample (including, importantly, in samples of non-UK born subjects) and strict controls for multiple statistical comparisons are used. Sex differences were examined. Limitations of the study are acknowledged. I have only a few comments for the consideration of the authors:

1. Page 13, line 5, the authors note that they measured the overall thickness between the ELM and the inner and outer photoreceptor segments. I wanted to double check whether the authors meant the ELM or ILM (internal limiting membrane), because the ELM is adjacent to the inner segments, and also because the distance between the ILM and photoreceptors is sometimes used a variable in OCT studies.
2. The section beginning on page 12 -- Genetic correlation and heritability enrichment patterns – is interesting but reports correlations with data from outside the UK Biobank sample. Since the paper is already long and with many complex findings and methods, the authors should consider either moving this section to a supplemental document, or perhaps reporting those findings in a separate paper.
3. The same comment applies to the following section -- Genetic causal links with brain disorders, where data from the FinnGenn database are used.
4. Regarding this section on genetic causal links with brain disorders: Even though the analyses suggest causality in a statistical sense, isn't it possible that effects observed in the retina and in the brain are both manifestations of the same underlying process, and that a discussion of causality doesn't really capture what is actually happening biologically? If this section is going to be kept in the paper, some further discussion of the nature of the relationship – beyond causality as implied from constrained statistical analyses – would be helpful.
5. The Discussion is brief relative to the rest of the paper. That's not necessarily a problem, but the authors could consider saying more about potential clinical applications of their findings, not only in psychiatric and neurological practice, but in routine health care for the population in general (including in children).

Response to the Reviewer 1:

Thank you so much for your careful review and constructive suggestions! Based on your suggestions, we have made substantial improvements and updated our manuscript accordingly. Here, we provide our responses to each of your points. For your convenience, we first state your comments in *italic* and then provide our responses.

Zhao et al has looked into retinal imaging biomarkers and compared to the brain structure and function modalities. They examined images from the UK biobank (fundus and OCT) (n=6,454 NHW individuals) with a full GWAS for those traits, and then correlated those with MRI modalities from brain. Associations were found for 121 tests that 66 replicated in an independent dataset. This information shows the clear relationship between eye and brain phenotypes, as well as the association between eye imaging features and brain, both of which can be used to diagnose disorders. Heritability was also determined for the eye features found in OCT. The independent datasets were in non-European multi ancestry datasets (n=6,490) as well as those of European ancestry who were not British (n=5320). There is a huge association of different genetics with both brain and eye diseases, and the overlap in subsequent features that can then correlate with multiple diseases or phenotypes. This study is extremely interesting and shows a large overlap between those phenotypes, imaging modalities, and genetics that has not been previously shown. This work is of significance and novel, with fascinating methodology.

Response: Thank you very much for reviewing our paper and for your supportive comments! We totally agree with the significance of the eye-brain connections identified in this study through imaging genetics. We have tried our best to improve this work by closely following your minor questions below.

Comments:

Some minor questions for clarity:

(1) How genetically did you determine the European ancestry? PCs were adjusted for but is there a category in the UKB that indicates those specific ancestries? (Page 35 details)

Response:

Thank you for pointing this out! Yes, the UKB database has a variable (“Data-Field 21000” <https://biobank.ndph.ox.ac.uk/showcase/field.cgi?id=21000>) for ethnic background and its accuracy has been verified using genotyping data in Bycroft et al. (*Nature*, 2018, PMID: 30305743) and was used widely in previous studies. We have clarified this in the revised manuscript:

“The ancestry information was derived from self-reported ethnicity (Data-Field 21000), the accuracy of which was confirmed in Bycroft, et al. ¹⁵¹.” (Page 37, Lines 25-27)

(2) The UKB participants have disease phenotypes. If those were not excluded, then of course different eye features or brain features would be found to correlate with disease genetics or with each other since those phenotypes have multiple features known to be included. How did you exclude/account for those? Some phenotypes were included (Page 34 lines 11-16) and accounted for, but not for any ocular or brain disorder?

Response: Thanks a lot for your comments and questions! The UKB participants have disease phenotypes, yet the cohort is generally healthier than the wider population, not specifically targeting any diseases. This trend, wherein UKB participants are found to be healthier than the general population (e.g., PMID: 28641372), results in a lower-than-expected number of cases at the population level. This phenomenon is even more pronounced among the subset of UKB participants in the imaging study, as individuals with serious diseases are less likely to participate in MRI data collection. Consequently, in our analysis, we did not exclude subjects based on the presence of ocular or brain disorders, aligning with the approach of many previous imaging-based UKB studies.

In revising this work, we conducted a sensitivity analysis following your suggestions, incorporating considerations for ocular or brain disorders. We began by retrieving the complete ICD-10 codes (“Data-Field 41270”, <https://biobank.ndph.ox.ac.uk/ukb/field.cgi?id=41270>) from the UKB study, identifying all cases of ocular or brain disorders characterized by ICD-10 codes starting with “F” (Mental and behavioural disorders), “G” (Diseases of the nervous system), or “H00-H59” (Diseases of the eye and adnexa). When we matched these conditions with subjects who had both retinal and brain images, we found that, among 219 diseases, 55 had more than 50 cases each (with an average of 62 cases, a median of 22 cases, and a range from 2 to 852 cases). Notably, the three conditions exceeding 500 cases were all related to cataracts (H25, H269, and H26). We then reran our phenotype association analysis between 156 retinal imaging traits and 458 brain MRI traits, including these 55 diseases as additional covariates. **Figure R1** (R = response letter) illustrates a comparison of regression coefficients before and after adjusting for these disease covariates. The analysis revealed consistent patterns with our main findings across different categories of retinal and brain images, indicating that disease status does not substantially alter the patterns observed in our main analysis. This sensitivity analysis has been detailed in the Supplementary Note and is briefly mentioned in the main text:

*“We performed additional sensitivity analysis to account for the ocular and brain disorders as covariates. We found consistent patterns with those observed in our main analysis, more details can be found in the **Supplementary Note (Figs. S11-12)**.”* (Page 7, Lines 12-15)

We have conducted a sensitivity check in our genetic analysis too, finding a high level of consistency when adjusting for ocular and brain disorder status. Further details are provided in our response to your **Comment (4)** below.

Figure R1: Phenotypic eye-brain associations in sensitivity analyses. The correlation coefficients between retinal imaging traits and brain MRI traits adjusting for ocular and brain disorders (y-axis) and not (x-axis). Results are separately displayed for OCT measures (upper panels) and fundus image traits (lower panels).

(3) How did you handle the multi-ancestry in the UKB for the independent dataset? Was sub ancestry analysis performed?

Response: Thank you for your question! As mentioned in our response to your **Comment (1)**, we used “Data-Field 21000” from the UKB database to gather ancestry information, selecting all subjects of British European descent as our discovery sample, while the remaining subjects were used for replication analysis (after removing their relatives). To maximize sample size while maintaining a certain level of specificity in this replication data analysis, we divided subjects with diverse ancestry backgrounds into two groups: European (excluding British, average $n = 5,320$) and non-European (average $n = 6,490$). For both groups, we corrected for population stratification by adjusting for the top 40 genetic principal components (PCs). We have provided more detailed explanations in the paper:

“Relatives of the discovery GWAS sample were removed, and we adjusted for the same set of covariates as in the discovery sample, including the top 40 genetic PCs to account for population stratifications.” (Page 37, Lines 27-29)

Therefore, for the non-European multi-ancestry subjects, we did not further separate them due to the small sample size, choosing instead to adjust for genetic PCs. However, inspired by your questions, we conducted separate analyses for subjects of Asian (average $n = 1,937$) and Black (average $n = 1,989$) ancestries—the two largest sub-cohorts—in the revised manuscript. This analysis enabled us to provide more precise ancestry-specific replication data, despite the much reduced sample sizes and statistical power. We have briefly mentioned this analysis in the main text and provided comprehensive details in the Supplementary Note, along with updates in Table S5.

“We also performed the replications analysis separately for UKB subjects of Asian and Black ancestries (Supplementary Note).” (Page 9, Lines 4-6)

(4) It seems to be obvious that if you included those with retinal disease, the ocular features would indicate glaucoma (for example) and be correlated significantly (lines 20-27 etc page 9) – so were those samples separated out based on disease phenotype?

Response: Thank you for your insightful comments! As mentioned in our response to your **Comment (2)**, the UKB cohort predominantly consists of healthy individuals, and we did not exclude subjects based on the presence of retinal or brain diseases. We have examined the overlap with all retinal or brain diseases as mentioned above and confirmed that disease status does not substantially impact the phenotypic associations.

Similarly, we performed a sensitivity analysis for our genetic study on all the 156 retinal imaging traits (covering both OCT and fundus images). We accounted for retinal or brain diseases as additional covariates. We focused on 88 of the 219 diseases, each with over 300 cases in the retinal imaging cohort (average case count was 644, median was 234, and the range spanned from 52 to 7,641). Notably, the three conditions with over 5000 cases were all related to cataracts

(H25, H269, and H26). We then re-conducted our GWAS for the 156 retinal imaging traits, incorporating these 88 diseases as covariates. After obtaining the GWAS summary statistics, we used LD score regression (LDSC) to assess the genome-wide genetic correlation for each trait, before and after disease status adjustment. The LDSC genetic correlation estimates remained remarkably consistent, with values ranging between [0.9985, 1.0001] for OCT images and [0.9982, 1.0014] for fundus images, suggesting minimal impact of disease status on the discovered genetic underpinnings of retinal imaging traits. This implies that the genetic correlations observed between retinal imaging phenotypes and ocular diseases are more likely due to the imaging' relevance to these conditions. We have provided this sensitivity analysis in the Supplementary Note and provided a brief overview in the main text:

*“Sensitivity analysis, which additionally included ocular and brain disorders as covariates, yielded highly consistent GWAS results (**Supplementary Note**).”* (Page 8, Lines 9-11)

(5) What is the hold-out independent dataset (n=959) – were these from both ancestry types? How were these determined? (Page 34 line 20)

Response: Thank you for your questions! Indeed, we used all individuals of white British ancestry from the UKB study as our discovery cohort. For the independent hold-out dataset, we included available subjects from all other ancestry backgrounds, excluding relatives up to the third degree of the discovery cohort as identified in Bycroft et al. (*Nature*, 2018, PMID: 30305743). We have enhanced the clarity of these procedures in the methods section:

“This hold-out dataset included all subjects not used in the discovery dataset, encompassing various ancestry backgrounds. Additionally, relatives¹⁵¹ of individuals from the discovery sample were excluded.” (Page 36, Lines 16-18)

(6) Why were top 10 PCs used for the non-European subjects instead of the 40? Due to ancestry? (Page 35 line 23-24)

Response: Thank you very much for pointing it out! Initially, our rationale for using a smaller number of genetic PCs was due to the much smaller sample size of non-European replication dataset, where 10 PCs are generally sufficient to capture population structures. However, upon re-examination, we found that we actually used all 40 PCs in this eye-brain project, aligning with the covariates taken for the discovery cohort. We have accordingly revised our descriptions to reflect this:

“..... and we adjusted for the same set of covariates as in the discovery sample, including the top 40 genetic PCs to account for population stratifications.” (Page 37, Lines 27-29)

(7) Why was validation performed on the UKB European subjects without retinal imaging data? Did they have MRI data? (Page 35 line 28-29)

Response: Thank you for your question! The PRS-CS method requires a validation dataset as input for estimating the LD structures of the genetic variants. Typically, this involves using an independent dataset separate from the GWAS discovery dataset. Thus, we select European subjects from the UKB outside of the retinal imaging cohort, ensuring they are distinct from the discovery sample yet representative of the LD patterns among genetic variants. Consequently, these subjects may or may not have brain MRI data, as the only requirement is having no retinal imaging data. We have revised our descriptions to better reflect this point:

“To avoid overlapping with the training discovery GWAS sample, the validation genotype data were randomly selected from 1,500 UKB European subjects without retinal imaging data.” (Page 37, Line 32 - Page 38, Line 2)

(8) Why was HapMap 3 instead of 1000 genomes variants used for the LD scores? (Page 37 line2)

Response: Thank you for your question! We obtained the precalculated LD scores from the LDSC website, which were derived from the 1000 Genomes Project samples but limited to the high-quality HapMap 3 variants (and removed the MHC region), as detailed in the LD Score Estimation Tutorial on their GitHub page (<https://github.com/bulik/ldsc/wiki/LD-Score-Estimation-Tutorial>). This approach follows the default recommendations and setup advised by the original LDSC methodology. Intuitively, using a curated selection of high-quality genetic variants in the LD scores may lead to reduced standard errors, as it excludes redundant genetic variants. We have enhanced the clarity of our explanations in the methods section:

“The default European LD scores provided by the LDSC software were used, which were based on the HapMap3 genetic variants and were estimated from the 1000 Genomes European samples. The major histocompatibility complex region was excluded due to the complex LD pattern.” (Page 39, Lines 4-7)

(9) How were the samples separated for training, validation and testing (Page 38 lines 14-16) from the entire UKB dataset?

Response: Thanks a lot for pointing it out! We used a two-step procedure to separate the subjects. First, we randomly allocated unrelated white British subjects, who had both retinal and brain MRI images, into validation and testing datasets. This strategy was designed to enable assessing the combined predictive capabilities of retinal imaging and brain MRI in later steps, which requires that the validation and testing datasets include subjects with both types of imaging. Second, all the remaining subjects with retinal imaging were used as the training dataset. We have made this process clearer in the methods section:

“We focused on unrelated white British subjects and first randomly divided the subjects who had both retinal and brain imaging data into validation and testing datasets (2,464 subjects in each dataset). This design enabled us to test the prediction performance using both imaging types in

later steps. The remaining 50,944 subjects with retinal imaging data (but no brain MRI data) were used as the training dataset.” (Page 40, Lines 18-23)

(10) Could the sex specific analysis have been weighted by disease that are found more predominantly in females than in males (for example the fundus image traits were the ones that had found extra associations)? (Page 6 lines 25-27).

Response: Thank you very much for your insightful suggestions! We first analyzed the distribution of disease status across the two gender groups. Among the 55 diseases with over 50 cases each (as identified in response to your **Comment (2)**), the average male-to-female case ratio was 1.158 (with a median of 1.029 and a range from 0.310 to 3.944, illustrated in **Figure R2**). Subsequently, we considered these diseases as additional covariates in our sex-specific phenotypic analyses. Similar to the analyses with combined samples (**Figure R1**), we found consistent patterns before and after the adjustment for these diseases (**Figure R3**). We have provided these results in the Supplementary Note.

Figure R2: Ratio of male to female case rates across the 55 diseases with more than 50 cases among subjects having both retinal and brain imaging traits. Diseases are color-coded according to their ICD-10 Chapters: F for Mental and Behavioral Disorders, G for Diseases of the Nervous System, and H for Diseases of the Eye and Adnexa.

Figure R3: Phenotypic sex-specific eye-brain associations in sensitivity analyses. The correlation coefficients between retinal imaging traits and brain MRI traits adjusting for ocular and brain disorders (y-axis) and not (x-axis). Results are separately displayed for females (upper panels) and males (lower panels).

(11) How were the 142 regions determined (Page 8 line 21-23) and the 140 regions? Are they the same regions?

Response: Thank you for pointing this out! The regions in your question are the same and constitute a subset of the significant genomic loci ($P < 3.20 \times 10^{-10}$, that is, $5 \times 10^{-8}/156$) identified in the discovery cohort using FUMA. To define the LD boundaries, FUMA initially pinpointed independent significant variants, which were characterized by P -values below 3.20×10^{-10} and independence from other significant variants ($LD\ r^2 < 0.6$). Subsequently, FUMA established LD blocks by associating all variants (with $MAF \geq 0.0005$, including those from the 1000 Genomes reference panel) in LD ($r^2 \geq 0.6$) with at least one independent significant variant. Among these, FUMA recognized independent lead variants as those independent from one another ($LD\ r^2 < 0.1$). For independent significant variants in proximity to each other (<250 kb, as determined by their LD boundaries), FUMA consolidated their LD blocks into a single genomic region if they were adjacent. In the replication GWAS samples, we examined the replication of these regions identified and defined in discovery GWAS. We have provided clarification on this point in the Methods section:

“Briefly, FUMA first labeled independent significant variants, which were variants whose P values were smaller than 3.20×10^{-10} and were independent of other significant variants ($LD\ r^2 < 0.6$). Next, LD boundaries were defined by tagging all variants ($MAF \geq 0.0005$, including those from the 1000 Genomes reference panel) in LD ($r^2 \geq 0.6$) with at least one of these independent significant variants. Based on the independent significant variants, FUMA further defined independent lead variants as those that were independent from others ($LD\ r^2 < 0.1$). If there were independent significant variants that were close with each other (<250 kb based on their LD boundaries), FUMA merged their LD blocks into one single genomic locus to obtain the LD boundaries. For independently significant associations and LD blocks defined by FUMA, we performed validations using.....” (Page 37, Lines 13-23)

Response to the Reviewer 2:

Thank you so much for your careful review and constructive suggestions! Based on your suggestions, we have made substantial improvements and updated our manuscript accordingly. Here, we provide our responses to each of your points. For your convenience, we first state your comments in *italic* and then provide our responses.

The authors aim to make eye-brain-genetic connections using the UKB to determine which structures and genes are related and how they might inform brain disorders. While the study is well designed, the execution of the results and the discussion are a major limitation. It may be worth describing some of the more significant findings and saving space to discuss them, as compared to forgoing on a proper discussion of these very important results. For example, some very nice results relating to schizophrenia came up, but they are not discussed at all.

Response: Thank you very much for your constructive suggestions! We have substantially expanded the discussion of the results. We have also tried our best to address your other specific comments below.

Comments:

(1) The sentence should be reformatted "The retina, the only part of the central nervous system that can be visualized without surgical intervention, plays a critical role in the visual pathway." since it is not true that the brain cannot be visualized. This is done with MRI imaging for example. So it isn't clear why "surgical intervention" is mentioned. It is the case that retinal imaging is less invasive and provides greater detail than what is possible with brain MRI.

Response: Thanks a lot for pointing this out and detailed explanations! We have reformatted this sentence as following:

"The retina, an important component of the central nervous system that can be non-invasively visualized through retinal imaging, plays a critical role in the visual pathway." (Page 3, Lines 1-2)

(2) would consider deleting the word altered in line 8 "As a result, the retina serves as a unique window into altered brain structure/function1,3....."

Response: Thank you for the suggestion! It makes sense and we have accordingly removed this word!

(3) Were only healthy-ish people used for the analysis? If yes or no, please update line 12 where it states "We explored the genetic relationship between the eye and brain by analyzing multimodal retinal and brain imaging traits from the UK Biobank (UKB) study43.

Response: Thank you for your question! The UKB study primarily consists of a healthy cohort, hence we did not exclude samples based on their disease history. Following insights and suggestions from the Reviewer 1, we conducted a series of sensitivity analyses to assess the

impact of eye and brain diseases on our results. Generally, we observed that the disease case numbers were low, and that findings of our phenotypic and genetic analyses remained consistent whether or not these diseases were taken into account. Further details are provided in our response to Reviewer 1's Comments (2, 4, and 10) and in the Supplementary Note of the paper. Following your suggestions, we have revised this sentence to enhance clarity:

"We explored the genetic relationship between the eye and brain by analyzing multimodal retinal and brain imaging traits from the UK Biobank (UKB) study⁴³. The majority of our study's cohort consists of healthy subjects, with an overall low prevalence of ocular or brain disorders (Supplementary Note)." (Page 4, Lines 12-15)

(4) What retinal thickness measures were included? This would be important to note given their relationships between retinal layers and layers of the brain.

Response: Thank you for pointing this out! We thought your question was related to this sentence *"The OCT-derived measurements, such as retinal thickness across layers, and vertical cup-to-disc ratio, were already available in the UKB database."*

In this project, we analyzed 43 measures of retinal thickness, which were derived from several previous studies. A comprehensive list of these measures is provided in Table S1. To enhance clarity in this revision, we have included detailed descriptions within Table S1 and incorporated specific examples in the revised sentence:

"The OCT derived measures, including retinal thickness across layers^{44,45} (such as the retinal nerve fiber layer [RNFL], inner nuclear layer [INL], and the ganglion cell and inner plexiform layer [GCIPL]) and vertical cup-to-disc ratio³⁰, were already available in the UKB database." (Page 4, Lines 16-19)

(5) Figure 3, some of the labels for A should be updated to reflect the correct nomenclature for the retinal layers. For example, what is ELM_ISOS_thickness?

Response: Thank you for your careful review! For the OCT traits, we closely adhered to their provided names in the UKB database. For example, the ID "ELM_ISOS_thickness_left" corresponds to the official name "Average ELM-ISOS thickness (left)" found at <https://biobank.ndph.ox.ac.uk/showcase/label.cgi?id=100079>, under Data-Field ID 28520. This label denotes the mean thickness spanning the ELM (external limiting membrane) layer and the ISOS (inner and outer photoreceptor segments). To enhance understanding through visualization, we have indicated the locations of several layers, including the ELM and ISOS, in Figure 3B.

To enhance the readability of Figure 3A, we have revised the labels for clarity following your suggestions. For example, the ID "ELM_ISOS_thickness_left" has been changed to "Thickness between ELM and ISOS (left eye)". Furthermore, we have clarified in the figure legend that ELM stands for external limiting membrane and ISOS denotes the inner and outer photoreceptor segments.

(6) Figure 2, not sure why B and C start with describing DTI measures when in A, the analysis starts with volume. Please try to make the ordering consistent. I would make D, E, and F (volume and thickness findings) go first and then follow that up by B and C (DTI findings)

Response: Thanks a lot! Following your suggestions, we have prioritized the presentation of volume and thickness findings, proceeding then to the DTI findings. Accordingly, we have updated Figure 2, reorganized the Supplementary Figures, and adjusted the order of these parts in the main text.

(7) Why weren't Visual cortical regions extracted from the brain MRI measures. There are 2 tools built into FreeSurfer that can do this and it would have provided more accurate information related to V1 through V5/MT, as well as fusiform gyrus sub-regions, which have provided a more detailed map of these associations.

Response: Thank you for these suggestions! The brain structural MRI data used in this study were produced by previous projects (e.g., PMIDs 31676860 and 37262162), which used ANTS (<https://stnava.github.io/ANTs/>) for region delineation and volume extraction, instead of FreeSurfer.

In this revision, we have examined the volumetric traits generated using FreeSurfer. We searched and found that the UKB imaging team has provided the following related phenotypes on visual cortical regions:

Freesurfer BA exvivo (<https://biobank.ndph.ox.ac.uk/showcase/label.cgi?id=195>)

- 27098 Volume of MT (left hemisphere)
- 27140 Volume of MT (right hemisphere)
- 27096 Volume of V1 (left hemisphere)
- 27138 Volume of V1 (right hemisphere)
- 27097 Volume of V2 (left hemisphere)
- 27139 Volume of V2 (right hemisphere)

Regional and tissue volume FAST (<https://biobank.ndph.ox.ac.uk/showcase/label.cgi?id=1101>)

- 25854 Volume of grey matter in Temporal Fusiform Cortex, anterior division (left)
- 25855 Volume of grey matter in Temporal Fusiform Cortex, anterior division (right)
- 25856 Volume of grey matter in Temporal Fusiform Cortex, posterior division (left)
- 25857 Volume of grey matter in Temporal Fusiform Cortex, posterior division (right)
- 25858 Volume of grey matter in Temporal Occipital Fusiform Cortex (left)
- 25859 Volume of grey matter in Temporal Occipital Fusiform Cortex (right)
- 25860 Volume of grey matter in Occipital Fusiform Gyrus (left)
- 25861 Volume of grey matter in Occipital Fusiform Gyrus (right)

Therefore, we downloaded these 14 traits and conducted analyses with retinal imaging traits. We found that FreeSurfer traits also showed significant associations with retinal imaging traits, exhibiting overall patterns similar to those observed with RNFL, GCIPL, and macular thickness.

For example, strong associations were found between V1 and both GCIPL and overall macular thickness measurements, paralleling the associations found with pericalcarine volume using ANTs. Additionally, we identified new associations, such as the volume of grey matter in the occipital fusiform gyrus, which not only correlated with GCIPL and macular thickness measures but also with subfields between the INL and RPE layers, as well as between the INL and ELM layers. We have briefly mentioned this analysis in the main text and detailed the results in the Supplementary Text:

*“These regional brain volume traits were extracted using advanced normalization tools (ANTs)⁴⁸, we additionally conducted further analyses with traits of visual cortical regions generated from Freesurfer⁴⁹, which yielded consistent findings (**Supplementary Note and Fig. S4**).”* (Page 6, Lines 5-8)

Figure R4: Phenotypic associations between OCT measures and brain structural MRI traits from Freesurfer. We illustrate correlations between derived OCT measures (y-axis) and regional brain volumes generated by Freesurfer (x-axis). The color represents correlation estimates. The asterisks indicate significant correlations at the FDR-significance level ($P < 1.25 \times 10^{-5}$). The five numbers in the names are their UKB Data-Field IDs. Based on these IDs, more information about these traits can be found at <https://biobank.ndph.ox.ac.uk/showcase/label.cgi?id=195> or <https://biobank.ndph.ox.ac.uk/showcase/label.cgi?id=1101>.

(8) Fix "excitatory"

Response: Thank you for your careful review! We have fixed this.

(9) The following sentence should be fixed for great clarity "As neuroprotective treatments become more widely available, this ability to predict brain diseases could have major clinical benefits."

Response: Thank you so much for your suggestions! We have revised this sentence as follows:

"Specifically, the discovery of retinal imaging phenotypes and their shared genetic risk factors with brain disorders could pave the way for less invasive monitoring and early detection methods, providing valuable insights into the pathological processes of these conditions." (Page 18, Lines 6-9)

(10) Too much time is spent describing the results and very little time spent discussing what they might mean. there are much greater implications than what is currently discussed.

Response: Thank you for your comments and suggestions! We have diligently worked to expand the Discussion Section by focusing on the following areas: 1) schizophrenia and other neuropsychiatric disorders; 2) neurological disorders; 3) glioma/glioblastoma; and 4) the potential of routine health care for brain health in the general population. These discussions are primarily located on Pages 18-19.

(11) The following sentence needs further clarifying so that it makes sense to readers who are not familiar with UKB data "Furthermore, inferring phenotypic causality from our current cross-sectional analysis is challenging." In particular I am concerned about the inferences being made regarding diagnosis or other traits. For example, in the UK biobank, a diagnosis could have been identified either before or after imaging was performed.

Response: Thank you very much for your insightful questions and suggestions! The mentioned sentence is related to retinal and brain imaging phenotypes (rather than diagnoses), for which we presently have data from only a single observation time point per subject. We aim to emphasize that, as repeated retinal and brain imaging data become available in the future (for example, <https://pubmed.ncbi.nlm.nih.gov/37355273/>), it will enable us to trace the longitudinal trajectory of changes in the eye-brain axis, potentially shedding light on underlying causal relationships. We have made revisions to this sentence accordingly:

"Furthermore, inferring phenotypic causality between retinal and brain imaging phenotypes from our current cross-sectional analysis is challenging. Repeated UKB eye and brain imaging scans¹³⁷ in the future will enable us to study the causal relationships between eye and brain structural and functional changes using a longitudinal study design." (Page 19, Lines 14-17)

Response to the Reviewer 3:

Thank you so much for your careful review and constructive suggestions! Based on your suggestions, we have made substantial improvements and updated our manuscript accordingly. Here, we provide our responses to each of your points. For your convenience, we first state your comments in *italic* and then provide our responses.

Zhao et al. describe a set of analyses, mainly based on the UK Biobank sample, examining the overlap between the genetic contributions to eye and brain structure. Specifically, they examined correlations between genetic findings from 156 retinal imaging variables (derived from color fundus photography and OCT), and 458 brain imaging variables (derived from structural MRI, DTI, and fMRI (resting state and task-based)). This is a timely and important paper, given the accumulating evidence that changes in the eye, especially in the retina, can concurrently or longitudinally predict changes in brain structure, brain function, and disease status (e.g., from no disease to a range of cardiovascular and neurodegenerative diseases; or from milder to more severe disease). The findings from this paper confirm the genetic bases for these effects. In addition, they suggest specific brain changes that are linked to retinal changes, and the genes involved in those relationships. Some of these findings are novel, such as the relationships between retinal findings and white matter tract integrity. The paper should stimulate further research on eye-brain connections, eye-brain disease connections, and the genetics of these connections. The findings are interesting, and the paper overall is a groundbreaking report.

This paper has many strengths. It is well-written and easy to follow. Prior literature is comprehensively cited, methods and results are described in detail, and the discussion sticks closely to the findings. Results are validated outside of the model training sample (including, importantly, in samples of non-UK born subjects) and strict controls for multiple statistical comparisons are used. Sex differences were examined. Limitations of the study are acknowledged. I have only a few comments for the consideration of the authors:

Response: Thank you very much for your encouraging feedback! We have improved our paper based on your comments below.

Comments:

(1) Page 13, line 5, the authors note that they measured the overall thickness between the ELM and the inner and outer photoreceptor segments. I wanted to double check whether the authors meant the ELM or ILM (internal limiting membrane), because the ELM is adjacent to the inner segments, and also because the distance between the ILM and photoreceptors is sometimes used a variable in OCT studies.

Response: Thank you for your suggestions and explanations! The measure referred to as the “overall thickness between the ELM and the inner and outer photoreceptor segments” is original from the Data-Fields 28520 (<https://biobank.ndph.ox.ac.uk/showcase/field.cgi?id=28520>) and 28521 (for the left and right eyes, respectively). The official description of Data-Field 28520 states

it as “Average thickness measured between the external limiting membrane (ELM) to the inner and outer photoreceptor segments (ISOS) across all subfields in the left eye.” This phenotype was initially generated based on research found at <https://pubmed.ncbi.nlm.nih.gov/27720551/>. Furthermore, the locations of ELM and ISOS are illustrated in Figure 3B of our main text.

Upon review, we found that the UKB database does not currently provide data on the distance between the ILM and the photoreceptors among the OCT traits. The available variables related to the ILM are Data-Fields 28542 (<https://biobank.ndph.ox.ac.uk/showcase/field.cgi?id=28542>, “QC - ILM indicator (left eye)”) and 28543 (“QC - ILM indicator (right eye)”), which were used in our analysis as quality control measures, following recommendations from the literature.

(2) The section beginning on page 12 -- Genetic correlation and heritability enrichment patterns – is interesting but reports correlations with data from outside the UK Biobank sample. Since the paper is already long and with many complex findings and methods, the authors should consider either moving this section to a supplemental document, or perhaps reporting those findings in a separate paper.

Response: Thank you so much for your suggestions! Following the communications and suggestions from the Editor, we have chosen to keep this section in the paper. Given that the genetic correlation and heritability enrichment analyses typically only need summary-level GWAS summary statistics data, they facilitate the integrating GWAS results of different complex traits and diseases collected from different sources. Consequently, investigating genetic correlation patterns across different studies has emerged as a popular approach. For example, the main method we used, LD score regression (LDSC, <https://pubmed.ncbi.nlm.nih.gov/26414676/>), has been cited in over 3,000 publications, predominantly in analyses that span multiple cohorts.

(3) The same comment applies to the following section -- Genetic causal links with brain disorders, where data from the FinnGenn database are used.

Response: Thank you for your suggestions! In line with the Editor’s suggestions, we have kept this section. The investigation of genetic causal relationships was conducted using two-sample Mendelian randomization (MR) methods, which intrinsically depend on GWAS summary statistics from separate study cohorts. Hence, the choice of a UK Biobank – FinnGen design for our analysis was a fitting approach.

(4) Regarding this section on genetic causal links with brain disorders: Even though the analyses suggest causality in a statistical sense, isn't it possible that effects observed in the retina and in the brain are both manifestations of the same underlying process, and that a discussion of causality doesn't really capture what is actually happening biologically? If this section is going to be kept in the paper, some further discussion of the nature of the relationship – beyond causality as implied from constrained statistical analyses – would be helpful.

Response: Thank you for your insightful comments! Indeed, the two-sample MR approach we used serves as an epidemiological tool for causal inference. We agree that it may not yield in-

depth biological insights into the underlying complex mechanisms, given that it uses GWAS genetic variants as instrumental variables. Following your advice, we have enriched the Discussion section with more discussion of this limitation and constraint inherent to the MR approach:

“In addition, the two-sample MR method used in this study, an epidemiological approach for genetic causal inference, relies on assumptions on genetic variants and may not offer full insights into inter-organ biological processes. Insights at the molecular and cellular levels are needed to better understand the connections between the eye and the brain¹³⁸.” (Page 19, Lines 18-22)

(5) The Discussion is brief relative to the rest of the paper. That’s not necessarily a problem, but the authors could consider saying more about potential clinical applications of their findings, not only in psychiatric and neurological practice, but in routine health care for the population in general (including in children).

Response: Many thanks for your detailed suggestions! Following your comments, we have expanded the Discussion Section on these areas: 1) schizophrenia and other neuropsychiatric disorders; 2) neurological disorders; 3) glioma/glioblastoma; and 4) routine health care for brain health in the general population. These discussions are primarily located on Pages 18-19.

REVIEWERS' COMMENTS

Reviewer #1 (Remarks to the Author):

The authors have most satisfactorily resolved all the questions I had in the last review, particularly the ones relating to ancestry, genetics, and the covariates like sex and disease that could have impacted their analysis, but did not, thanks to the significant information added and analyses performed by the authors.

Reviewer #2 (Remarks to the Author):

The authors did an excellent job addressing all of the comments made by reviewers. I am satisfied with the resubmitted manuscript.

Reviewer #3 (Remarks to the Author):

The authors have done an excellent job revising the manuscript based on the comments of the three reviewers of the earlier version. I believe the manuscript is now stronger, and is acceptable for publication in this form.